# Combining Recurrent Neural Networks with Variational Mode Decomposition and Multifractals to Predict Rainfall Time Series

Hai Zhou, Daniel Schertzer, and Ioulia Tchiguirinskaia

Hydrology Meteorology & Complexity (HM&Co), Ecole des Ponts ParisTech, Champs-sur-Marne, France

**Correspondence:** Hai Zhou (hai.zhou@enpc.fr)

**Abstract.** Rainfall time series prediction is essential for monitoring urban hydrological systems, but it is challenging and complex due to the extreme variability of rainfall. A hybrid deep learning model (VMD-RNN) is used in order to improve prediction performance. In this study, variational mode decomposition (VMD) is first applied to decompose the original rainfall time series into several sub-sequences according to the frequency domain, where the number of decomposed sub-sequences is determined by power spectral density (PSD) analysis. To prevent the disclosure of forthcoming data, non-training time series are sequentially appended for generating the decomposed testing samples. Following that, different recurrent neural network (RNN) variant models are used to predict individual sub-sequences and the final prediction is reconstructed by summing the prediction results of sub-sequences. These RNN-variants are long short-term memory (LSTM), gated recurrent unit (GRU), bidirectional LSTM (BiLSTM) and bidirectional GRU (BiGRU), which are optimal for sequence prediction. In addition to three common evaluation criteria, mean absolute error (MAE), root mean square error (RMSE) and mean absolute percentage error (MAPE), the framework of universal multifractals (UM) is also introduced to assess the performance of predictions, which enables the extreme variability of predicted rainfall time series to be characterized. The study employs two rainfall time series with daily and hourly resolutions, respectively. The results indicate that the hybrid VMD-RNN model provides a reliable one-step-ahead prediction, with better performance in predicting high and low values than the pure LSTM model without decomposition.

## 1 Introduction

Prediction of rainfall time series plays an important role in monitoring urban hydrological systems and their geophysical environment. Accurate and trustworthy predictions can serve as an early warning of floods and other extreme events, as well as a guide for water resource allocation. Although predicting rainfall time series is not a novel concept, it has remained fundamentally difficult due to the extreme variability, in fact intermittency, of rainfall over a wide range of space-time scales, i.e. increasingly heavy precipitation is concentrated over smaller and smaller fractions of the space-time.

Classical forecast models are either process-driven physical models or data-driven statistical models. The former represents the most important physical processes and numerically solves the governing equations based on initial and boundary conditions (Lynch, 2008). Due to the fact that rainfall depends on a variety of land, ocean, atmospheric processes, and their complex interactions, physical models are developed based on simplifications of those processes, in particular by truncating

the scales and introducing rather ad-hoc parametrisations. This greatly increases their unpredictability (Bauer et al., 2015). On the contrary, data-driven models strive to establish a link between input and output data to predict time series without regard to underlying physical processes (Reichstein et al., 2019). In general, they provide a unique output therefore with no information on the uncertainty generated by the nonlinearity of the involved processes. A sort of hybrid approach has been developed using stochastic models physically based on the cascade paradigm (e.g., Schertzer and Lovejoy, 1987; Marsan et al., 1996; Schertzer and Lovejoy, 2004, 2011). This ensures that intermittency is directly taken into account, including in the generation of uncertainty.

The explosion of supercomputing and data availability offers immense potential for data-driven models to significantly contribute to prediction (Schultz et al., 2021). There are several methods available for predicting rainfall time series, including linear and nonlinear models. The traditional linear data-driven model is the autoregressive integrated moving average (ARIMA) (Chattopadhyay and Chattopadhyay, 2010), which ignores the nonlinearity of the relationship between input and output time series, leading to poor prediction ability. Because of increased data availability and computing power, various deep learning (DL) models have been proposed and applied in predicting nonlinear time series (Lara-Benítez et al., 2021).

Recurrent neural network (RNN) models are a subset of deep learning models, which have been specifically designed to solve sequential prediction problems (Elman, 1990). However, standard RNN struggles with long-term dependence and exhibits the gradient vanishing or exploding problems (Hochreiter and Schmidhuber, 1997). RNN variants, such as long short-term memory (LSTM), gated recurrent unit (GRU), bidirectional LSTM (BiLSTM) and bidirectional GRU (BiGRU), are intended to alleviate the limitations of standard RNN. These variant models have been employed in various fields (e.g., Graves et al., 2013; Cho et al., 2014; Su et al., 2020; Lin et al., 2022), including time series prediction (e.g., Ma et al., 2015; Ding et al., 2019; Gauch et al., 2021). In particular, great efforts have been devoted to predicting rainfall time series (e.g., Ni et al., 2020; Barrera-Animas et al., 2022; He et al., 2022), as shown in Table 1.

However, these pure variant models are not always capable of efficiently handling extremely nonlinear time series with several noisy components without the need for appropriate preprocessing (Liu et al., 2020; Huang et al., 2021; Zhang et al., 2021; Lv and Wang, 2022; Ruan et al., 2022). Decomposition is a typical preprocessing method in time series analysis, which can extract hidden information to aid in the comprehension of the complex original time series. For decomposition approaches, wavelet decomposition (Pati et al., 1993), empirical mode decomposition (EMD) (Huang et al., 1998) and variational mode decomposition (VMD) (Dragomiretskiy and Zosso, 2013) are commonly used to decompose original data. Relevant studies on time series prediction by combining decomposition technique with deep learning models are also presented in Table 1. Because wavelet decomposition is highly dependent on the choice of the mother wavelet function, its adaptability in decomposing time series is limited (Hadi and Tombul, 2018). Meanwhile, EMD suffers from boundary effects, mode mixing, and a lack of exact mathematical foundations (Devi et al., 2020). In comparison, VMD, which is theoretically sound, presents the advantage of solving the mode overlap problem.

The inherent variability of rainfall typically results in limited prediction performance for single RNN-variant models. In response to this situation, integrated forecasting paradigms have been widely employed to improve the precision and robustness of time series forecasting. The hybrid VMD-RNN model is based on the fundamental concept of considering the dominant

characteristics of VMD in decomposing nonlinear time series and the beneficial performance of variant RNN models in predicting complex sequential problems.

The main purpose of this study is to provide a reliable one-step-ahead rainfall prediction for hydrological applications, particularly urban flood forecasting and water resource management. This addresses the fundamental challenge in operational hydrology where accurate short-term precipitation forecasts are essential for timely flood warnings and infrastructure management. In order to achieve this objective, it is essential to fully extract the underlying patterns of rainfall time series while preserving their intermittency structure - a critical requirement for hydrological modeling where extreme events often dominate system response. An additional crucial point is to develop prediction models with a satisfactory level of accuracy for practical implementation in operational hydrological systems. According to the aforementioned two factors, this study implements a hybrid approach known as VMD-RNN, which combines different RNN-variant models with VMD decomposition for predicting rainfall time series.

The effectiveness and reliability of the employed VMD-RNN approach are extensively validated by applying this method to forecast the following step's rainfall in both daily and hourly resolution, representing different temporal scales relevant to hydrological practice. Furthermore, a comparison study is carried out to further demonstrate the superiority of the adopted VMD-RNN model, in comparison to the baseline method, the pure LSTM model without decomposition, and linear regression method. In addition, the UM technique is used to confirm the ability of the predicted time series to accurately describe rainfall variability, ensuring that the predicted series maintain the multifractal properties essential for accurate hydrological modeling and flood risk assessment.

Given the growing usage of deep learning in hydrological research, it is important to bridge the knowledge gap for readers who are not familiar with deep learning models. The pedagogical aspect of our work has the potential to contribute to the hydrology community by providing a deeper understanding of the application of deep learning models and multifractal techniques in short-term rainfall prediction that remains a fundamental problem of hydrology starting with one-step-ahead prediction. This work specifically addresses the need in the HESS community for accessible methodological advances that maintain strong connections to hydrological theory and practice, demonstrating how modern deep learning techniques can enhance traditional approaches to precipitation forecasting while preserving the physical understanding of rainfall processes essential for water resource management.

The rest of this article is organized as follows. In section 2, the corresponding methodologies are presented in detail, including VMD, RNN variants and UM. Two rainfall time series with daily and hourly resolutions are performed by VMD-RNN in section 3. The results are discussed and analyzed in section 4. Finally, conclusions and future work are given in section 5.

[Table 1 about here.]

## 2 Methodology

### 2.1 Variational mode decomposition

The primary process of variational mode decomposition (VMD) is constructing and solving the variational problem (Dragomiretskiy and Zosso, 2013). For rainfall time series $f(t)$, the variation problem is described as identifying $K$ sub-sequences $u_k(t)$ with center frequency $\omega_k$ to minimize the sum value of the estimated bandwidth of each $u_k(t)$. The constrained condition is that the aggregation of the sub-sequences $u_k(t)$ should be equal to the original sequence $f(t)$. The constrained variational problem can be expressed as follows:

$$\min_{\{u_k\},\{\omega_k\}} \left\{ \sum_{k=1}^{K} \left\| \partial_t \left[ \left( \delta(t) + \frac{j}{\pi t} \right) * u_k(t) \right] e^{-j\omega_k t} \right\|_2^2 \right\} \qquad \text{s.t.} \sum_{k=1}^{K} u_k = f(t) \tag{1}$$

where $\{u_k(t)\} = \{u_1(t), u_2(t), ..., u_K(t)\}$ and $\{\omega_k\} = \{\omega_1, \omega_2, ..., \omega_K\}$ are shorthand notations for decomposed sub-sequences and their center frequencies, respectively; $\delta(t)$ is the Dirac distribution, the symbol $*$ denotes convolution and $e^{-j\omega_k t}$ is a phasor describing the rotation of the complex signal in time, with $j^2 = -1$.

The variational problem is addressed efficiently using the alternate direction method of multipliers (ADMM). The modes $u_k(t)$ are updated by Wiener filtering in the Fourier domain with a filter tuned to the current center frequency, see Eq. (2), then the center frequencies $\omega_k$ are updated as the center of gravity of the corresponding mode's power spectrum, expressed as Eq. (3), and finally the Lagrangian multiplier $\lambda$ enforcing exact constraints is updated as the dual ascent by Eq. (4). The updating procedure is repeated until the convergence condition is satisfied, as in Eq. (5).

$$\hat{u}_k^{n+1}(\omega) \leftarrow \frac{\hat{f}(\omega) - \sum_{i<k} \hat{u}_i^{n+1}(\omega) - \sum_{i>k} \hat{u}_i^{n}(\omega) + \frac{\hat{\lambda}^n(\omega)}{2}}{1 + 2\theta(\omega - \omega_k^n)^2} \tag{2}$$

$$\omega_k^{n+1} \leftarrow \frac{\int_0^\infty \omega \left| \hat{u}_k^{n+1}(\omega) \right|^2 d\omega}{\int_0^\infty \left| \hat{u}_k^{n+1}(\omega) \right|^2 d\omega} \tag{3}$$

$$\hat{\lambda}^{n+1}(\omega) \leftarrow \hat{\lambda}^n(\omega) + \tau \left( \hat{f}(\omega) - \sum_k \hat{u}_k^{n+1}(\omega) \right) \tag{4}$$

$$\sum_k \frac{\left\| \hat{u}_k^{n+1} - \hat{u}_k^n \right\|_2^2}{\left\| \hat{u}_k^n \right\|_2^2} < \epsilon \tag{5}$$

where $\hat{u}_k^{n+1}(\omega), \hat{f}(\omega)$ and $\hat{\lambda}^{n+1}(\omega)$ represent the Fourier Transforms of $u_k^{n+1}(t), f(t)$ and $\lambda^{n+1}(t)$, respectively; $n$ is the iterations, $\theta$ is a quadratic penalty term, $\tau$ is the iterative factor that indicates VMD's noise tolerance and $\epsilon$ denotes the convergence tolerance.

## 2.2 Recurrent neural network

Recurrent neural network models perform deep learning by a unique recurrent structure (Elman, 1990), as illustrated in Figure 1. In terms of time series predicting, the recurrent units remember earlier information, processing not only new data but also previous outputs to generate an up-to-date prediction. However, RNN models have difficulty dealing with long-term information. Additionally, standard RNN suffers from the gradient vanishing or exploding problem. To overcome the constraints of standard RNN, long short-term memory (LSTM), gated recurrent unit (GRU), bidirectional LSTM (BiLSTM) and bidirectional 120 GRU (BiGRU), these variants of RNN are designed. Their working principles are explained in detail as follows.

[Figure 1 about here.]

### 2.2.1 Long short-term memory

LSTM models are explicitly constructed with special recurrent structures to remember information for long periods, and they have three gates to control the cell state that stores and conveys information (Hochreiter and Schmidhuber, 1997), which is 125 depicted as Figure 2. The forget gate $f_t$ determines how much information should be forgotten from the cell state, which constructs the long-term memory, as represented in Eq. (6). The input gate $i_t$ is responsible for deciding what new information should be stored in the cell, and the corresponding equations are Eq. (7) and Eq. (8). The output gate $o_t$ is to generate outputs, Eq. (9), and update the cell states $C_t$ and the hidden states $h_t$, expressed as Eq. (10) and Eq. (11) respectively.

[Figure 2 about here.]

$$f_t = \sigma\left(W_{xf}x_t + W_{hf}h_{t-1} + b_f\right) \tag{6}$$

$$i_t = \sigma\left(W_{xi}x_t + W_{hi}h_{t-1} + b_i\right) \tag{7}$$

$$\tilde{C}_t = \tanh\left(W_{xC}x_t + W_{hC}h_{t-1} + b_C\right) \tag{8}$$

$$o_t = \sigma\left(W_{xo}x_t + W_{ho}h_{t-1} + b_o\right) \tag{9}$$

$$C_t = f_t \otimes C_{t-1} + i_t \otimes \tilde{C}_t \tag{10}$$

$$h_t = o_t \otimes \tanh(C_t) \tag{11}$$

where $\sigma$ and tanh are activation functions, denoting sigmoid function and hyperbolic tangent function, respectively; $x_t$ is the input and $\tilde{C}_t$ is candidate memory; $W_{xf}$, $W_{xi}$, $W_{xC}$, $W_{xo}$ and $W_{hf}$, $W_{hi}$, $W_{hC}$, $W_{ho}$ represent the corresponding weights to $x_t$ and $h_{t-1}$; $b_f$, $b_i$, $b_C$ and $b_o$ are the related bias vectors; $\otimes$ indicates element-wise multiplication.

### 2.2.2  Gated recurrent unit

GRU also overcomes the drawbacks of standard RNN. Unlike LSTM, however, it only has two gates: a reset gate and an update gate (Cho et al., 2014). The rest gate $r_t$ is accountable for the short-term dependencies by determining which historical data should be forgotten, represented as Eq. (12). The update gate $z_t$ manages the long-term dependencies by controlling what information is delivered to the future, Eq. (13). The hidden state $h_t$ is then updated according to Eq. (4) and Eq. (15). The update gate performs functions similar to the forget and input gates of LSTM, so the recurrent structure of GRU (Figure 3) is less complex, which makes it more efficient computationally from a theoretical standpoint (Chung et al., 2014).

$$r_t = \sigma\left(W_{xr}x_t + W_{hr}h_{t-1} + b_r\right) \tag{12}$$

$$z_t = \sigma\left(W_{xz}x_t + W_{hz}h_{t-1} + b_z\right) \tag{13}$$

$$\hat{h}_t = tanh(W_{xh}x_t + W_{hh}(r_th_{t-1}) + b_h) \tag{14}$$

$$h_t = z_t \otimes h_{t-1} + (1 - z_t) \otimes \hat{h}_t \tag{15}$$

[Figure 3 about here.]

### 2.2.3  Bidirectional recurrent neural network

Bidirectional RNN (BiRNN) is an RNN variant model that takes into account both past and future information to predict the target (Schuster and Paliwal, 1997; Graves and Schmidhuber, 2005). The architecture of a bidirectional RNN is seen in Figure 4. It adds an additional hidden layer to the RNN construction so that information can be conveyed backward. The hidden state $h_t$ is obtained by concatenating the forward and backward hidden states, $\overrightarrow{h}_t$ and $\overleftarrow{h}_t$, implying that the output is generated by combining information from two hidden layers. To avoid the limitations of standard RNN, BiLSTM and BiGRU are used instead of BiRNN, which have excellent performance in time series prediction.

[Figure 4 about here.]

## 2.3 Universal multifractals

Universal multifractals (UM) have been widely used to describe nonlinear phenomena that have a multiplicative structure, such as rainfall. The core principle of the framework of UM is briefly explained here, and interested readers could refer to references (e.g., Schertzer and Lovejoy, 1987, 2011; Lovejoy and Schertzer, 2007) for more details. Let's denote $\varepsilon_\lambda$ is a conservative field at resolution $\lambda$ (=$L/l$, the ration between the outer scale of the phenomenon $L$ and the observation scale $l$), the statistical moment of order $q$ can be defined as:

$$\langle \varepsilon_\lambda^q \rangle \approx \lambda^{K(q)} \tag{16}$$

where $K(q)$ is the moment scaling function, characterizing the variability of the field at all scales.

In the UM framework, the moment scaling function $K(q)$ can be determined by two scale-invariant parameters $C_1$ and $\alpha$ in the conservative field, expressed as Eq. (17) (Schertzer and Lovejoy, 2011). $C_1$ is the mean intermittency co-dimension, which measures the average sparseness of the field. $\alpha$ is the multifractality index ($0 \leq \alpha \leq 2$), which indicates how fast the intermittency evolves when considering singularities slightly different from the average field singularity.

$$K(q) = \begin{cases} \frac{C_1}{\alpha-1}(q^\alpha - q) & \alpha \neq 1 \\ C_1 q \ln q & \alpha = 1 \end{cases} \tag{17}$$

The trace moment (TM) technique can be used to estimate UM parameters (Schertzer and Lovejoy, 2011; Gires et al., 2013). The steps in the technique are as follows: first, calculate the empirical statistical moment $\langle \varepsilon_\lambda^q \rangle$ (corresponding to the trace moment of fluxes) of order $q$ for each resolution $\lambda$, then plot the logarithm of the average field $\langle \varepsilon_\lambda^q \rangle$ versus the logarithm of $\lambda$, later perform linear regression to obtain the slope $K(q)$, and finally, according to the theoretical expression of $K(q)$ (Eq. (17)), $C_1$ is given by $K'(1) = C_1$ and $\alpha$ by $K''(1) = \alpha C_1$ because $\langle \varepsilon_\lambda^q \rangle = 1$, i.e. $K(1) = 0$ for the conservative field.

An alternative method for directly estimating the UM parameters $C_1$ and $\alpha$ is the double trace moment (DTM) (Lavallée et al., 1993; Gires et al., 2012). Based on the assumption that the conservative field $\varepsilon_\lambda^{(\eta)} = \frac{\varepsilon_\lambda^\eta}{\langle \varepsilon_\lambda^\eta \rangle}$ is renormalized by upscaling the $\eta$-power of the field at maximum resolution. Then, the statistical moment $K(q, \eta)$ of order $q$ is defined as: $\left\langle \varepsilon_\lambda^{(\eta)q} \right\rangle \approx \lambda^{K(q,\eta)}$ with $K(q,\eta) = K(q\eta) - \eta K(q)$. In the specific framework of UM, the statistical moment $K(q,\eta)$ can be expressed as $K(q,\eta) = \eta^\alpha K(q)$. Therefore, UM parameters $C_1$ and $\alpha$ are obtained according to the slope and intercept of the linear portion of the log-log plot $K(q,\eta)$ vs $\eta$.

When a multifractal field $\phi_\lambda$ is non-conservative ($\langle \phi_\lambda \rangle \neq 1$), it is usually assumed that it can be written as:

$$\phi_\lambda = \varepsilon_\lambda \lambda^{-H} \tag{18}$$

where $\varepsilon_\lambda$ is a conservative field ($\langle \varepsilon_\lambda \rangle = 1$) of the moment scaling function $K_c(q)$ depending only on $C_1$ and $\alpha$; $H$ is the non-conservation parameter ($H = 0$ for the conservative field).

The moment scaling function $K(q)$ of $\phi_\lambda$ is given by:

$$K(q) = K_c(q) - Hq \tag{19}$$

$H$ can be estimated using the following formula Tessier et al. (1993) :

$$\beta = 1 + 2H - K_c(2) \tag{20}$$

where $\beta$ is the spectral slope that characterizes the power spectrum of a scaling field, which follows a power law over a wide range of wave numbers:

$$E(k) \propto k^{-\beta} \tag{21}$$

Theoretically, a fractional integration of order $H$ (equivalent to a multiplication by $k^H$ in the Fourier space) is performed to retrieve $\varepsilon_\lambda$ from $\phi_\lambda$. A common approximation is to take $\varepsilon_\Lambda$ as the absolute value of the fluctuation of $\phi_\Lambda$ at the maximum resolution of $\Lambda$ and renormalizing it, shown as Eq. (22) in the one-dimension (Lavallée et al., 1993). Then, $\varepsilon_\lambda$ is obtained by upscaling $\varepsilon_\Lambda$.

$$\varepsilon_\Lambda = \frac{|\phi_\Lambda(i+1) - \phi_\Lambda(i)|}{\langle|\phi_\Lambda(i+1) - \phi_\Lambda(i)|\rangle} \tag{22}$$

## 3 Case study

### 3.1 Study area and datasets

Two rainfall time series with daily and hourly resolutions in Champs-sur-Marne (48.8425° N, 2.5886° E) were collected from MERRA-2 (Modern-Era Retrospective analysis for Research and Applications, Version 2) precipitation dataset that is produced by NASA's Global Modeling and Assimilation Office (GMAO), refer to The POWER Project (https://power.larc.nasa.gov). The corrected MERRA-2 precipitation dataset is a reanalysis product that integrates various observational data types (like radar, tipping bucket gauges, and satellite) through sophisticated data assimilation techniques into a climate model (Reichle et al., 2017).

One could worry about the model's applicability beyond the chosen study area, i.e., its transportability, because the model only has to be trained once. In principle, a new dataset from different regions or time periods can be fed directly into the well-trained model without repeating the training process to obtain the prediction on the new dataset.

The daily time series covered January 1, 2001 to December 31, 2020 (a total of 7305 data), of which from January 1, 2001 to January 7, 2015(5120 data, accounting for 70% of the total dataset) were selected as the training set while the remaining were

used as the non-training set. The non-training set was further divided into a validation set to tune hyperparameters according to loss changes and a testing set (1024 data, from March 14, 2018 to December 31, 2020) to evaluate the predicting performance, as presented in Figure 5(a). In addition, the rainfall time series with hourly resolution for the period between January 1, 2001 and November 1, 2001 (a total of 7305 data) was also studied and divided into three sets: a training set (5120 data), a validation set (1161 data), and a testing set (1024 data), as shown in Figure 5(b).

[Figure 5 about here.]

## 3.2 Model process

### 3.2.1 The implementation of VMD-RNN

In order to avoid using information from the future, the original rainfall time series was first divided into the training and non-training sets, and then the training set was decomposed into several sub-sequences and applied to train the models (Zhang et al., 2015; Zuo et al., 2020). To predict in the testing set, time series from the non-training set were sequentially appended to the training set, and the decomposition process was repeated with the rainfall time series of the next step appended. Following that, four variant RNN models were used to predict individual sub-sequences. The root mean square error (RMSE) was used to select the ideal RNN model with the optimal parameters for each sub-sequence. In addition to RMSE, UM was also employed to evaluate prediction performances, characterizing the extreme variability of time series. The implementation of the hybrid deep learning model (VMD-RNN) is summarized as follows and presented in Figure 6.

Step 1: Divide the original rainfall time series $f(t)$ ($t$=1,2,...,$N$, where $N$ is the length of total data) into a training set $f_T(t)$ ($t$=1,2,...,$N_t$, where $N_t$ is the training set length) and a non-training set $f_N(t)$ ($t$=1,2,...,$N_n$, where $N_n$ is the non-training set length).

Step 2: Use VMD to decompose the training set $f_T(t)$ into sub-sequences $u_{Ti}(t)$ ($i$=1,2,...,$K$).

Step 3: Sequentially append the non-training data $f_N(t)$ to the training set to generate $N_n$ new appended sequences $f_{NT}^j(t)$ ($j$=1,2,...,$N_n$ and $t$=1,2,...,$N_t+j$), and repeat decomposing each append sequence $f_{NT}^j(t)$ into $K$ sets of appended sub-sequences $u_{NTi}^j(t)$ ($i$=1,2,...,$K$).

Step 4: Exact the last sample $u_{NTi}^j(N_t+j)$ of each set of appended sub-sequences $u_{NTi}^j(t)$ as a non-training sample and divide the generated non-training samples $N_{vte}=N_n$ into two subsets: validation samples $N_v$ and testing samples $N_{te}$.

Step 5: For each sub-sequence, combine data from the training set and validation samples as history data, which is then used to train four variant RNN models and tune hyperparameters to find an ideal predicting model with optimal parameters.

Step 6: For each sub-sequence, input testing samples into the corresponding predicting models and obtain an individual predicted result $y_i(t)$ ($i$=1,2,...,$K$).

Step 7: Aggregate the predicted results of each sub-sequence to generate the final predicted result $y(t) = \sum_{i=k}^{K} y_i(t)$.

Step 8: Use the framework of UM to analyze the predicted and actual time series in the testing set.

To minimize the possibility of exposing future data during the decomposition of non-training time series, a precautionary approach (Step 3 and Step 4) has been implemented. This approach differs from the direct way of decomposing the testing time series using VMD. The non-training data was added to the training set in a sequential manner to create a new time series, and the amount of new generated time series was equal to the number of non-training data points. The VMD technique was thereafter used to decompose the aforementioned new time series into several sub-sequences. Subsequently, the final data point of each newly generated sub-sequence was retrieved and designated as non-training data, which was then used to build validation and testing samples.

[Figure 6 about here.]

### 3.2.2 Parameters of VMD

The decomposition performance of VMD is affected by the decomposition level $K$, the quadratic penalty term $\theta$, the convergence tolerance $\varepsilon$, and the noise tolerance $\tau$. In this study, the number of $K$ was identified by observing the power spectral density (PSD) of the last sub-sequence. The value of $K$ was determined on the training set with 5120 data. First, an initial $K$ value was given, such as $K = 5$, and there were five sub-sequences (IMFs) with the same length of training set. Then, each sub-sequence was divided into 40 samples with 128 data, to perform the spectral analysis and plot the corresponding PSD of sub-sequences. After that, $K$ was increased by one and the plotting PSD was repeated until the PSD of the last sub-sequence exhibited an evident change, compared with the previous last sub-sequence. For daily time series, the optimal number of $K$ was 8, which is depicted in Figure 7, whereas $K = 6$ for hourly time series. Based on the trial and error, other parameters of VMD were suggested as: $\theta$=100, $\varepsilon$=1e-9 and $\tau$=0.

[Figure 7 about here.]

### 3.2.3 Parameters of RNN

In the process of training, hyperparameters such as the number of inputs, epoch, hidden layers, and hidden units all influence the performance of models. Without loss of generality, the first sub-sequence (IMF1) is taken as an example to describe the determination of the ideal RNN structure with the optimal hyperparameters. The specific process is as follows: First, initial a single hidden layer model with 5, 10, 15 input neurons and 1 output neuron, and run different variants of RNN model (LSTM, GRU, BiLSTM, BiGRU) for various hidden neurons 32, 64 and 128. All experiments were intended to run for 10,000 epochs (one epoch is defined as when an entire dataset is passed forward and backward through the neural network only once), but early stopping with a large patience value (=200) was applied to prevent unnecessary overfitting, which means the model will stop the training if the performance on the validation dataset does not improve after 200 epochs. After adjusting hyperparameters, the ideal model with optimal parameters was found for the first sub-sequence (IMF1) where MAE and RMSE are the least.

The results of the model with one hidden layer for IMF1 predicting are shown in Table 2, where the best value is marked in bold. Then, different second hidden layers with hidden neurons 32, 64 and 128 were added to the first hidden layer with optimal parameters in order to discover the optimal parameters for the second hidden layer. By analogy, a third hidden layer was added. Table 3 shows the results of the optimal model with second and third hidden layers for IMF1 predicting. Through the above method, the variant RNN model structures of IMF1-IMF8 components were obtained, as shown in Table 4.

[Table 2 about here.]

[Table 3 about here.]

[Table 4 about here.]

### 3.3 Open-source software

This study made extensive use of open-source software. Python 3.8 was the programming language. The packages, Numpy (Van Der Walt et al., 2011), Pandas (McKinney et al., 2011), and Scikit-Learn (Pedregosa et al., 2011), were used to preprocess data. Tensorflow (Abadi et al., 2016) and Keras (Chollet et al., 2018) were the deep learning frameworks used to analyze time series, and Matplotlib (Hunter, 2007) was used to create all the resulting figures. The decomposition of time series by VMD was implemented based on the package of vmdpy (Carvalho et al., 2020), which is derived from the original VMD Matlab toolbox (Dragomiretskiy and Zosso, 2013). TM and DTM analysis were performed to calculate UM parameters according to the Multifractal toolbox that was provided by the website (https://hmco.enpc.fr/portfolio-archive/multifractals-toolbox) (Gires et al., 2011, 2012, 2013).

### 4 Result analysis

To verify the effectiveness of the hybrid VMD-RNN model, the benchmark methods, the pure LSTM model without decomposition and the linear regression (LR) method, were introduced. The benchmark also used the previous 5-day rainfall values to predict the next day's rainfall. The parameters for the pure LSTM were adjusted by trial and error. The qualitative and quantitative analysis of one-step-ahead predicted rainfall time series from two different models were conducted.

### 4.1 Daily rainfall series

Figure 8 shows the predicted daily time series in the testing set. It compares the predicted results of the VMD-RNN hybrid model, the pure LSTM model and the linear regression method with the actual data. It can be clearly observed that the hybrid model has a better fit for most of the points, particularly during periods of high-intensity rainfall events that are critical for flood forecasting applications. The VMD-RNN model demonstrates enhanced capability to capture rainfall variability patterns, including the temporal clustering of precipitation events that characterizes real rainfall processes.

The comparison of prediction performance with and without VMD for daily time series in the testing set can be seen in Figure 9. The scatter plot demonstrates that the VMD-RNN model has superior performance in predicting both high and low values for

daily time series, whereas the baseline models LSTM and linear regression exhibit systematic biases. Notably, the VMD-RNN model shows improved performance in predicting extreme rainfall events, which are crucial for urban flood warning systems.

The predicted values obtained by the baseline models exhibit considerable deviation from the best linear fitting line (blue dotted line), with a tendency to underestimate high-intensity events - a critical limitation for hydrological applications where accurate prediction of extreme events directly impacts flood risk assessment and emergency response effectiveness.

[Figure 8 about here.]

[Figure 9 about here.]

It was also necessary to know which model performed better from the quantitative aspect. Table 5 compares the results of three widely used criteria: RMSE, MAE, and MAPE. It can be seen that the three criteria of VMD-RNN are plainly lower, so the hybrid model outperforms the pure model. It further confirms the strong capability of the hybrid model in rainfall prediction.

[Table 5 about here.]

In addition to calculating the prediction error, the UM technique was also introduced to evaluate prediction performance

since it enables the extreme variability of rainfall time series to be characterized. According to Tessier et al. (1996), the rainfall series in France exhibits a rough scaling break phenomenon between 16 days and 30 days. Therefore, the analysis of UM starts with a range of scales from 1 day, increasing in powers of two to an outer scale of 16 days. Figure 10 presents $log \langle \varepsilon_\lambda^q \rangle$ versus $log \lambda$ over the range of $q$ between 0.3 and 2.5 with a coefficient of determination greater than 0.99, the log-log plot of $\left\langle \varepsilon_\lambda^{(\eta)q} \right\rangle$ vs $\lambda$ for $q = 1.5$, and the corresponding log-log plot of $K(q, \eta)$ vs $\eta$.

[Figure 10 about here.]

All the parameter values estimated using the TM and DTM methods are listed in Table 6. The values of $\alpha$ and $C_1$ obtained using the DTM technique show slight differences from those estimated by TM, but remain within acceptable ranges for multifractal analysis. Importantly, the VMD-RNN predicted time series preserves multifractal properties more effectively than LSTM without decomposition, as evidenced by UM parameters that are closer to those of actual rainfall. This preservation of

scaling properties is crucial for hydrological applications where the multifractal structure of rainfall directly influences runoff generation, infiltration processes, and the temporal distribution of streamflow in urban catchments.

[Table 6 about here.]

**4.2 Hourly rainfall series**

Figure 11 displays the hourly time series in the testing set with 1024 data points. The qualitative analysis reveals that the

predictive performance differences between VMD-RNN, pure LSTM, and linear regression are less pronounced for hourly rainfall time series compared to daily predictions. This reduced benefit of decomposition for hourly data can be attributed to

the inherently higher noise level and lower signal-to-noise ratio characteristic of high-frequency precipitation measurements, which limits the effectiveness of decomposition techniques in extracting meaningful frequency components.

Figure 12depicts the comparison between predicted and actual hourly rainfall values. The scatter plot reveals that the predicted values from VMD-RNN basically agree with the corresponding actual values, but the values predicted from the baseline LSTM model do not yield the same level of alignment. While the VMD-RNN model shows reasonable agreement with actual values for moderate to high rainfall intensities, significant challenges become apparent for low-intensity precipitation events.

[Figure 11 about here.]

[Figure 12 about here.]

The UM analysis results for hourly time series (Figure 13) and estimated parameters (Table 7) indicate that the predictive performance of VMD-RNN is comparable to pure LSTM for hourly data, without demonstrating the substantial benefits observed for daily predictions. The UM parameters $\alpha$ and $C_1$ show similar values between VMD-RNN and LSTM predictions, suggesting that both approaches preserve multifractal properties to a similar degree at hourly resolution. This finding reflects the scale-dependent effectiveness of decomposition techniques, where the benefits become more apparent at longer timescales where signal-to-noise ratios are higher and frequency separation is more pronounced.

[Table 7 about here.]

[Figure 13 about here.]

## 4.3 Discussion

### 4.3.1 Comparison with existing approaches

The hybrid VMD-RNN approach addresses several limitations of traditional hydrological forecasting methods. Compared to physically based numerical weather prediction models, data-driven approaches like VMD-RNN can provide more computationally efficient solutions for short-term rainfall prediction, particularly for local-scale applications where high-resolution atmospheric models may not be practical or cost-effective (Wilks, 2011). However, it is important to note that these models complement rather than replace physical understanding of hydrological processes.

Nowcasting systems, which typically rely on radar observations and numerical weather prediction models, face challenges in accurately predicting the timing and intensity of precipitation events (Hess and Boers, 2022). The VMD-RNN approach could potentially be integrated with existing nowcasting frameworks to improve short-term precipitation forecasts, particularly when combined with radar-based observations and ensemble forecasting techniques.

The multifractal analysis using Universal Multifractals provides additional insights into the scaling properties of rainfall that are not captured by traditional error metrics. The closer agreement of VMD-RNN predictions with observed multifractal parameters suggests that the model better preserves the natural variability structure of rainfall processes across scales (Schertzer et al., 1997). This is particularly important for hydrological applications where the temporal distribution and intensity patterns of rainfall can significantly affect runoff generation and flood risk.

### 4.3.2 Hydrological significance and applications

The results of this study demonstrate that the VMD-RNN hybrid approach offers significant advantages for rainfall prediction in hydrological contexts, particularly at daily time scales. This improvement has important implications for operational hydrology and water resource management.

The enhanced performance of VMD-RNN in capturing extreme rainfall events is particularly valuable for flood early warning systems. Extreme precipitation events are the primary drivers of flash floods and urban flooding, and their accurate prediction can provide crucial lead time for emergency response and flood mitigation measures (Berne et al., 2004; Beven, 2012). The improved prediction of high-intensity events could enhance the reliability of flood forecasting systems and reduce false alarm rates, which are critical factors in maintaining public trust and ensuring effective emergency response (Demeritt et al., 2007).

### 4.3.3 Model limitations and uncertainties

Despite the promising results, several limitations must be acknowledged. The systematic overestimation of low-intensity rainfall represents a significant challenge for practical applications. This bias could lead to overestimation of cumulative precipitation over extended periods, affecting water balance calculations and long-term hydrological planning (Gardiya Weligamage et al., 2023). The issue of false positives in low-intensity predictions is a common challenge in precipitation forecasting and requires careful consideration in operational applications.

The temporal scale dependency observed in our results, where VMD decomposition shows greater benefits for daily compared to hourly predictions, suggests that the approach may be most suitable for applications requiring daily to weekly rainfall forecasts. This scale dependence may be related to the frequency content of rainfall signals, where longer time scales contain more distinct frequency components that can be effectively separated by VMD.

The current study focuses on a single location with a temperate climate. The performance of VMD-RNN may vary significantly across different climatic regions, particularly in areas with distinct wet and dry seasons, monsoon climates, or arid regions where rainfall patterns differ markedly from those observed in our study area. Further validation across diverse climatic conditions is essential for establishing the general applicability of the approach.

## 5 Conclusions and future work

In this study, the hybrid VMD-RNN model was used as a methodology for forecasting rainfall with a one-step lead time. The integration of variational mode decomposition with recurrent neural networks demonstrates significant potential for improving rainfall time series prediction accuracy, particularly for extreme events that are critical for flood risk assessment.

VMD was first used to extract hidden information to understand the complex original time series. Then variants of RNN were applied to handle problems involving sequential prediction. By combining the dominant characteristics of VMD in decomposing nonlinear time series and the favourable performance of variant RNN models in predicting complex sequential problems,

the hybrid model based on VMD and RNN was employed to predict rainfall time series with daily and hourly resolution. The framework of UM was subsequently introduced to evaluate the performance of predicting rainfall time series.

According to the above study, the following conclusions could be drawn: (1) The VMD-RNN hybrid approach successfully addresses the challenge of predicting highly variable rainfall time series by decomposing the signal into frequency-specific components. The determination of optimal decomposition levels through power spectral density analysis provides a systematic approach for model configuration. (2) For daily rainfall prediction, the VMD-RNN model significantly outperforms pure LSTM models, particularly in capturing extreme rainfall events that are crucial for flood forecasting applications. The improvement in prediction accuracy has direct implications for early warning systems and flood risk management. (3) The closer agreement of VMD-RNN predictions with observed universal multifractal parameters demonstrates that the model better preserves the natural scaling variability of rainfall processes. This validation using $\alpha$ and $C_1$ parameters provide additional confidence in the model's ability to represent the complex intermittent nature of precipitation. (4) The benefits of VMD decomposition are more pronounced at daily compared to hourly time scales, suggesting that the approach may be most effective for applications requiring daily to weekly rainfall forecasts rather than sub-daily nowcasting.

However, there are still some limits to this study, and corresponding improvements will be implemented in future work. First, extending the approach to multi-step-ahead predictions would significantly enhance its practical utility for hydrological applications. Second, incorporating spatial information through the development of spatially distributed VMD-RNN models could improve rainfall prediction for catchment-scale applications. Third, the integration of physics-informed constraints into the VMD-RNN framework could help address some of the observed limitations, particularly the overestimation of low-intensity rainfall. Finally, the development of ensemble forecasting capabilities would provide valuable uncertainty information for decision-making.

*Code and data availability.* The source python code of VMD is available at https://github.com/vrcarva/vmdpy (Carvalho et al., 2020). The Multifractal toolbox is provided by the website (https://hmco.enpc.fr/portfolio-archive/multifractals-toolbox) (Gires et al., 2013, 2012, 2011). Two rainfall time series with daily and hourly resolutions in Champs-sur-Marne are collected from The POWER Project (https://power.larc.nasa.gov).

*Author contributions.* HZ, DS and IT developed the concept for the manuscript; HZ was responsible for conducting the data analysis, coding the hybrid model, and writing the first draft. DS and IT supervised the entire research and assisted with answering questions. All authors reviewed and edited the paper.

*Competing interests.* The authors declare that they have no conflicts of interest to report regarding the present study.

*Acknowledgements.* The first author is funded by the China Scholarship Council (202006120045). We would like to express our sincere gratitude to the editor and anonymous reviewers for their professional comments and helpful suggestions.

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

**List of Figures**

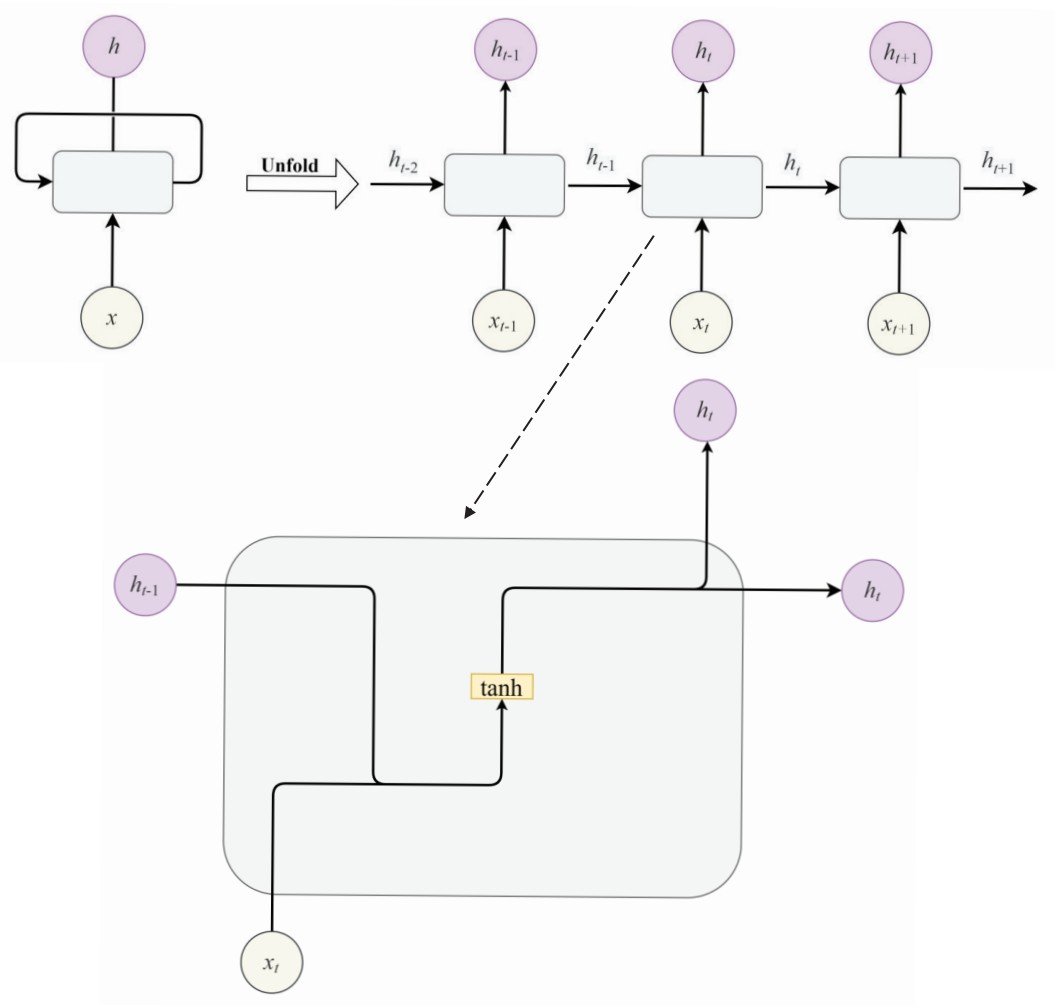

**Figure 1.** The structure of standard RNN

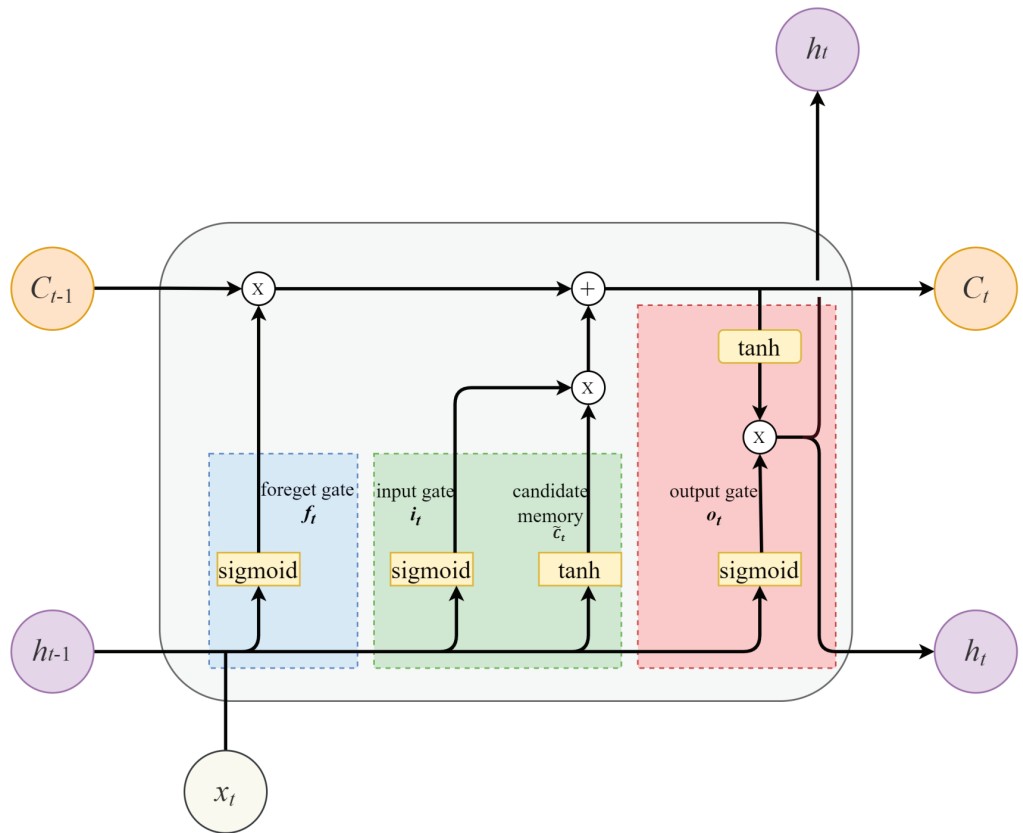

**Figure 2.** The recurrent structure of LSTM

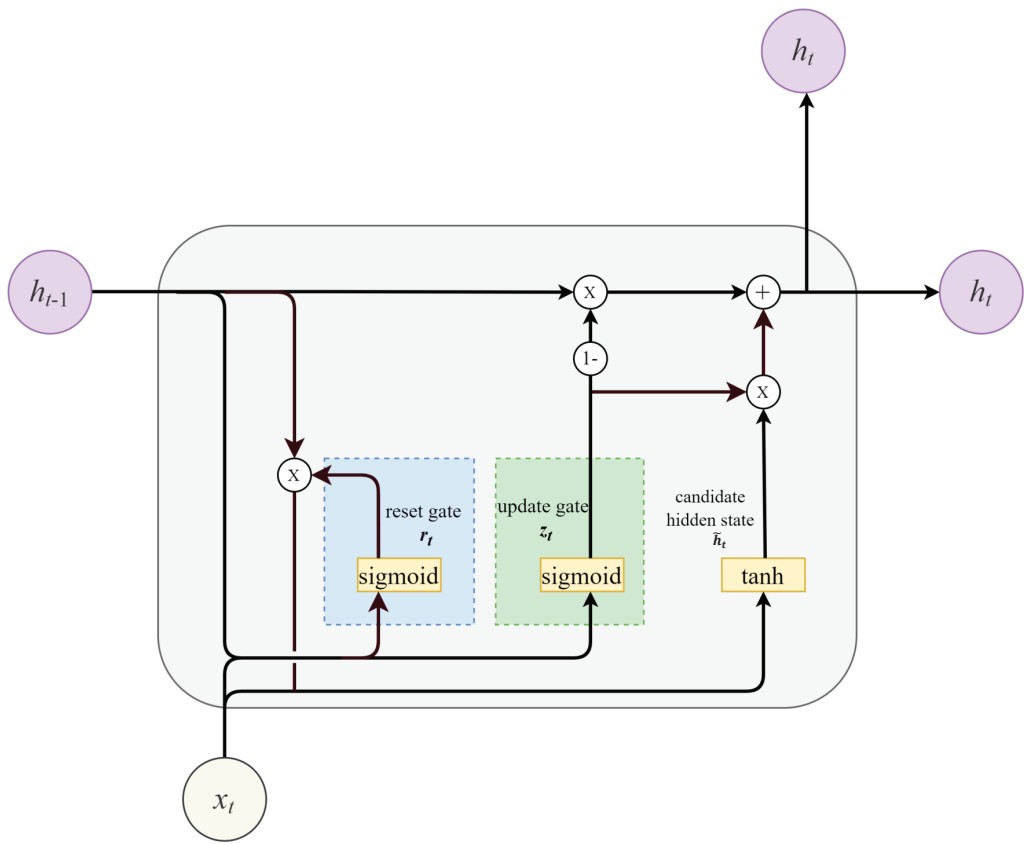

**Figure 3.** The recurrent structure of GRU

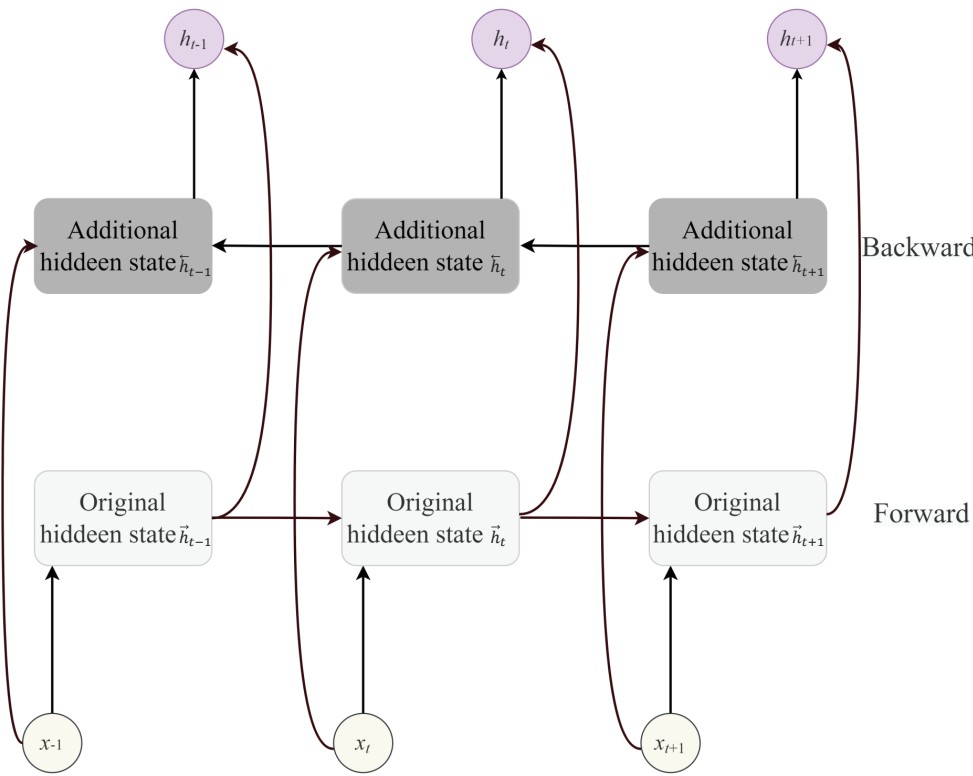

**Figure 4.** The structure of BiRNN

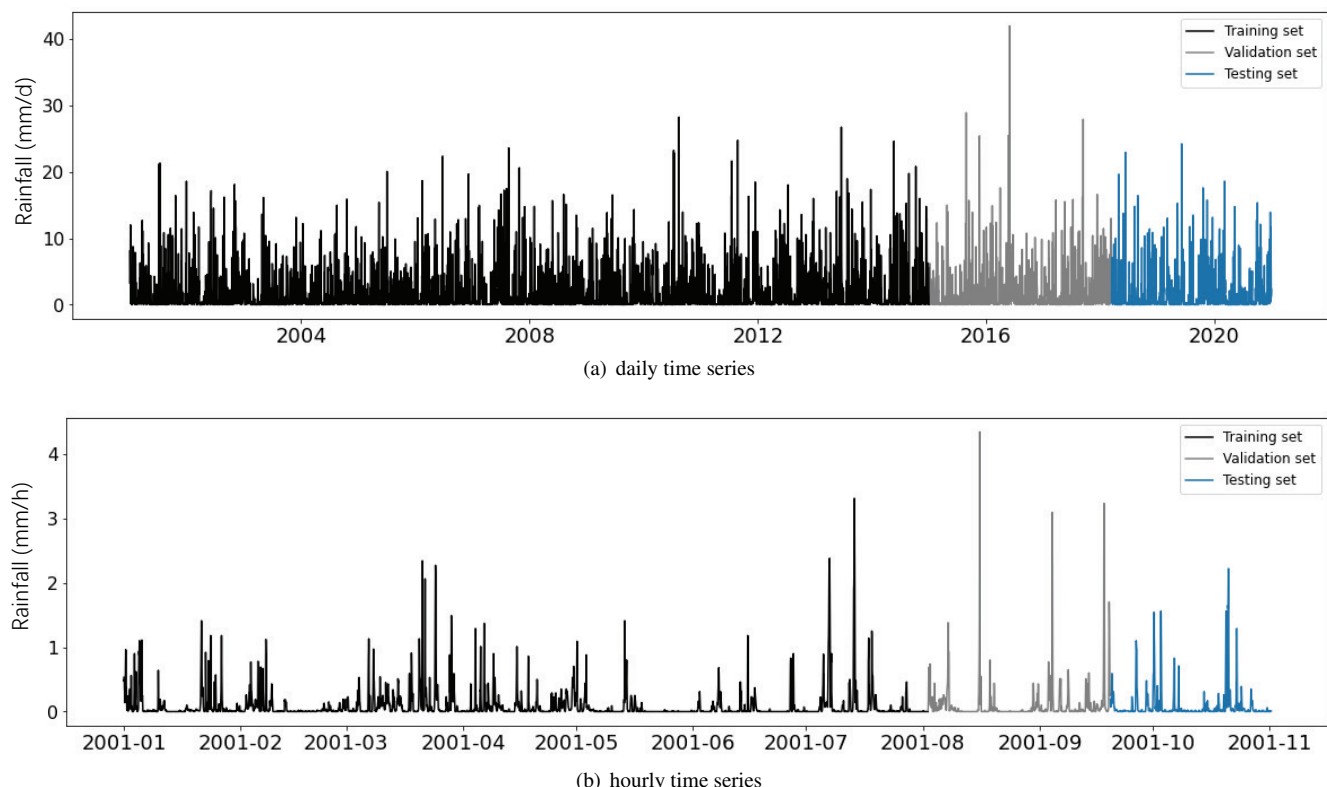

**Figure 5.** Original rainfall time series

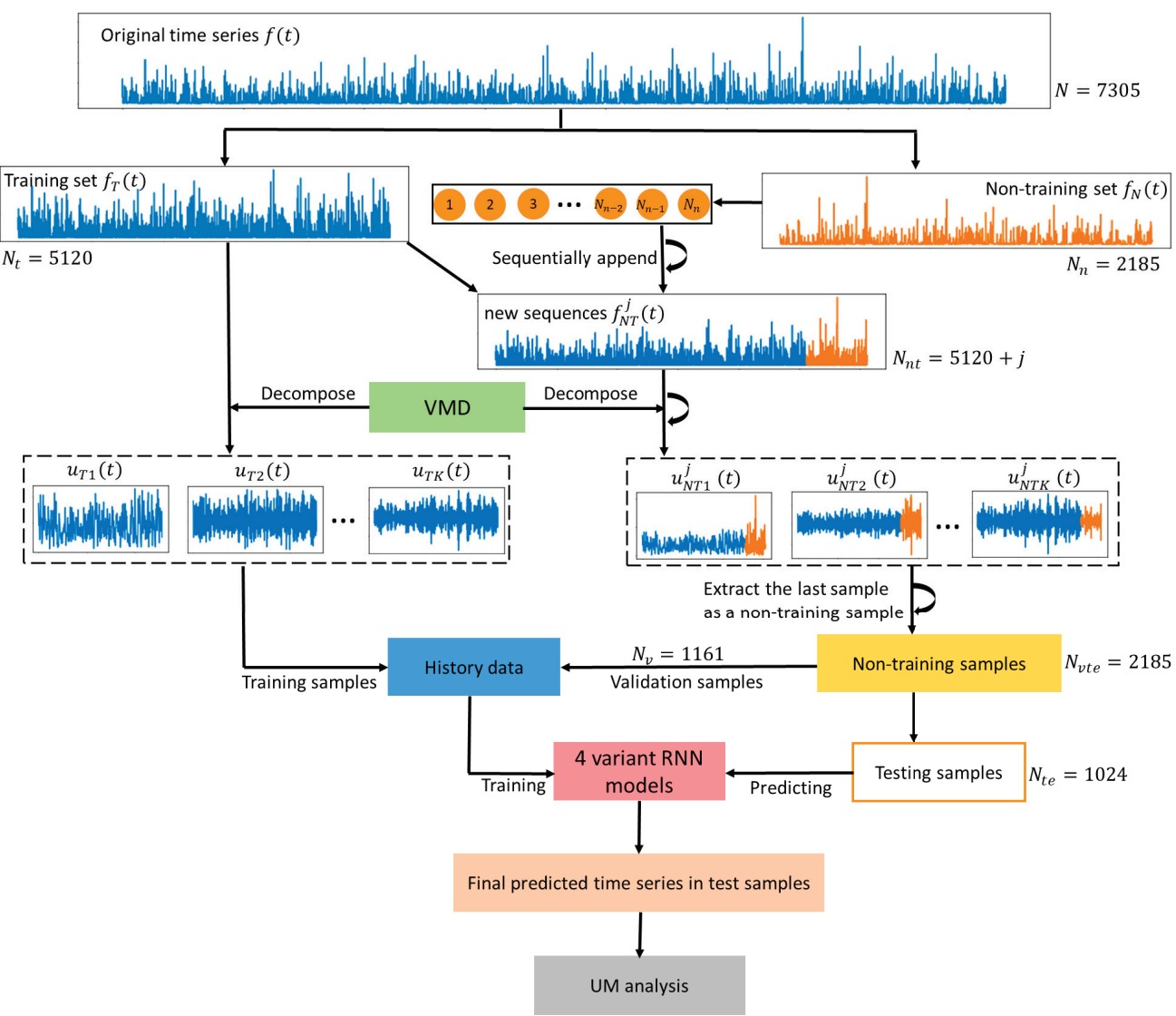

**Figure 6.** The process of the VMD-RNN model

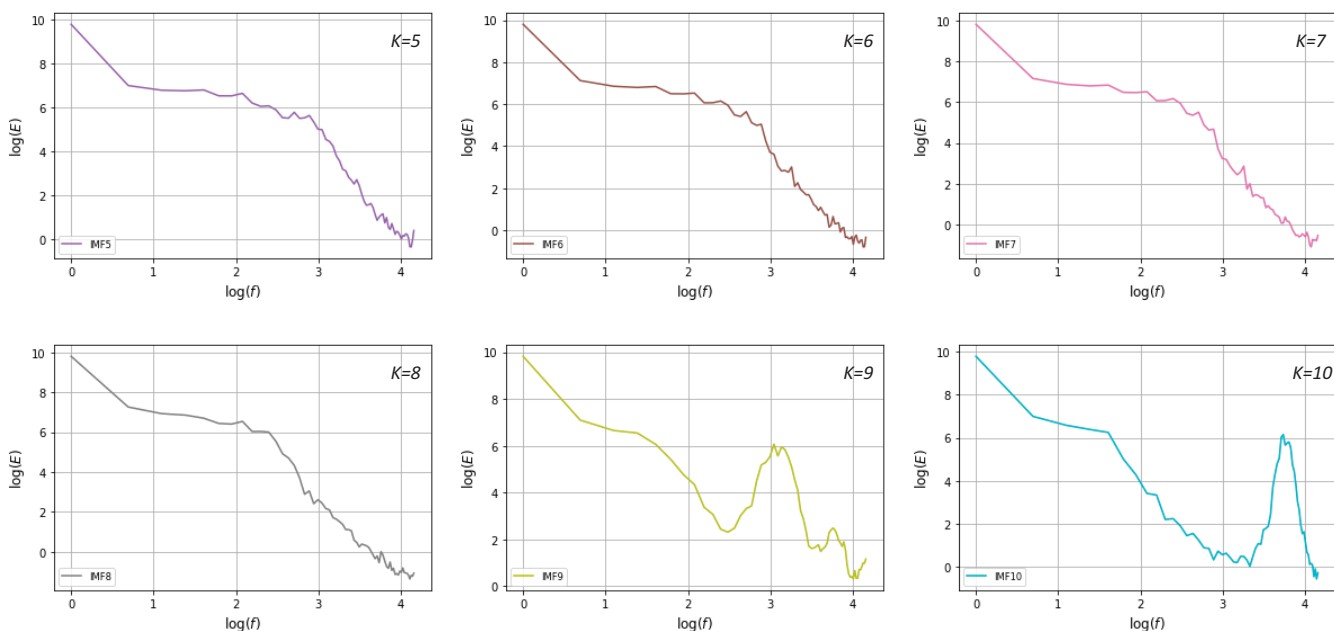

**Figure 7.** PSD of the corresponding last sub-sequence when K from 5 to 10

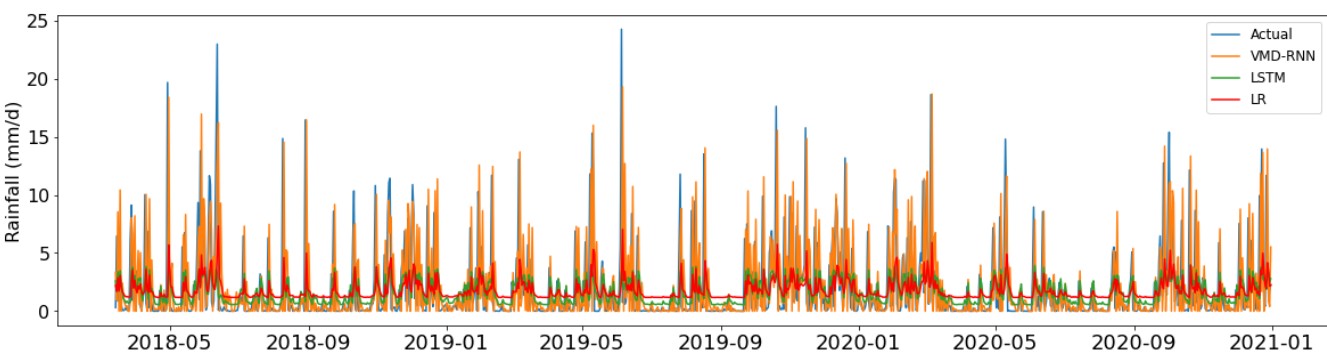

**Figure 8.** Predicted and actual daily time series in the testing set

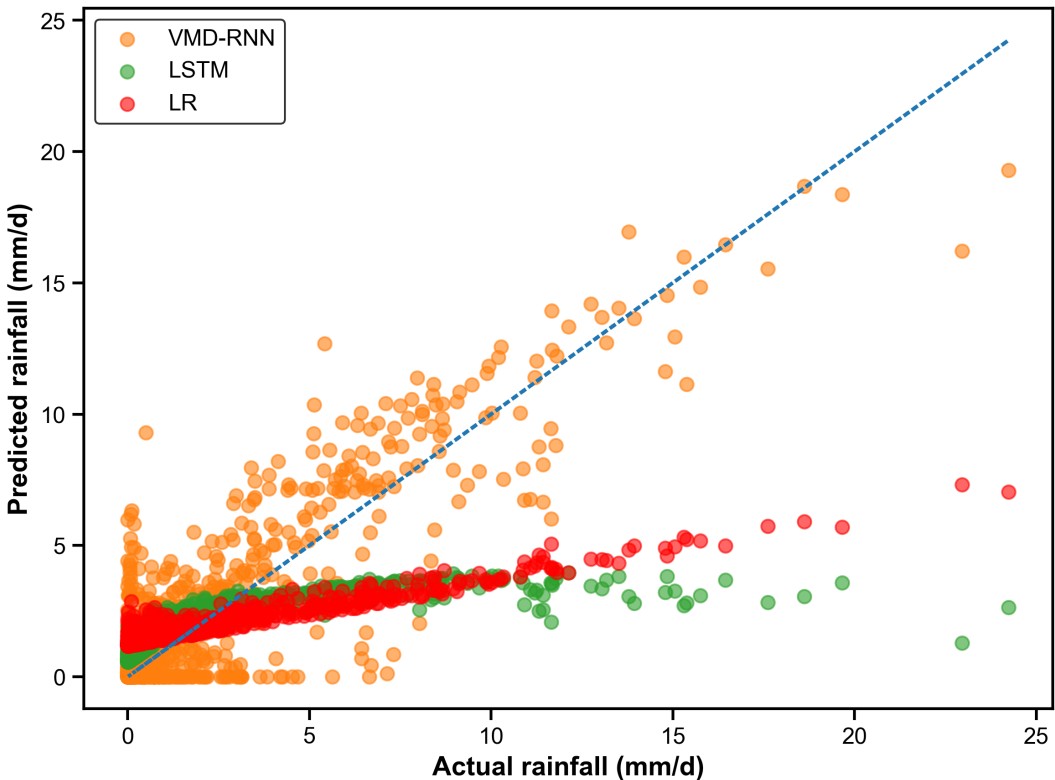

**Figure 9.** The comparison between predicted and actual daily rainfall values

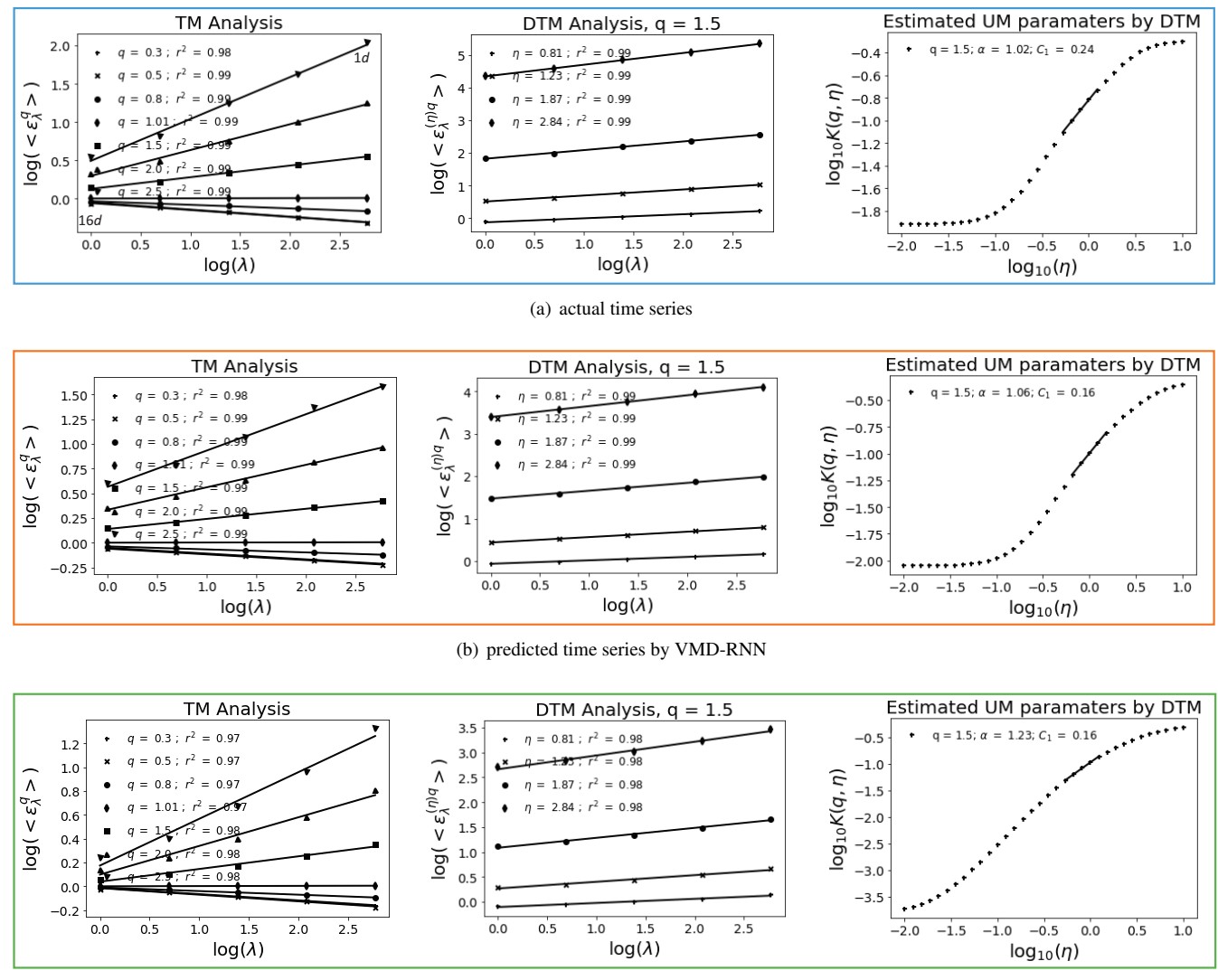

(a) actual time series

(b) predicted time series by VMD-RNN

(c) predicted time series by LSTM without decomposition

**Figure 10.** UM results for daily time series in the testing set

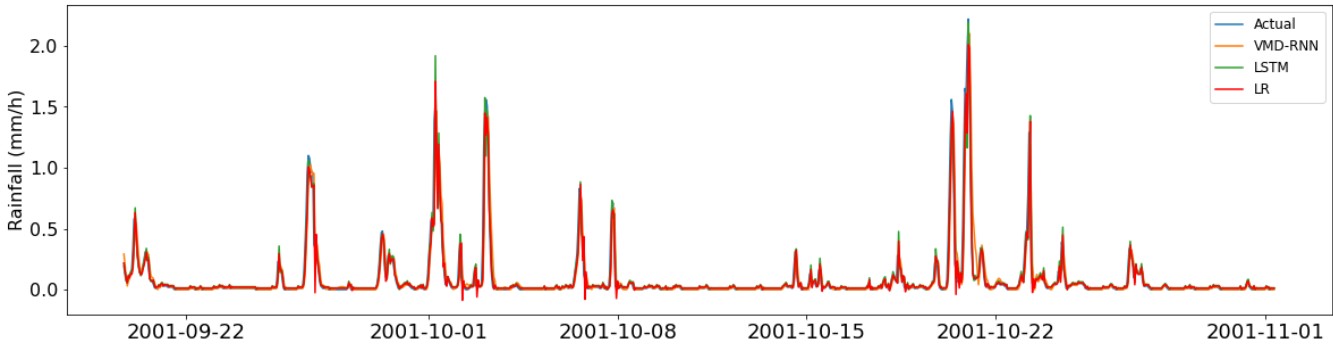

**Figure 11.** Predicted and actual hourly time series in the testing set

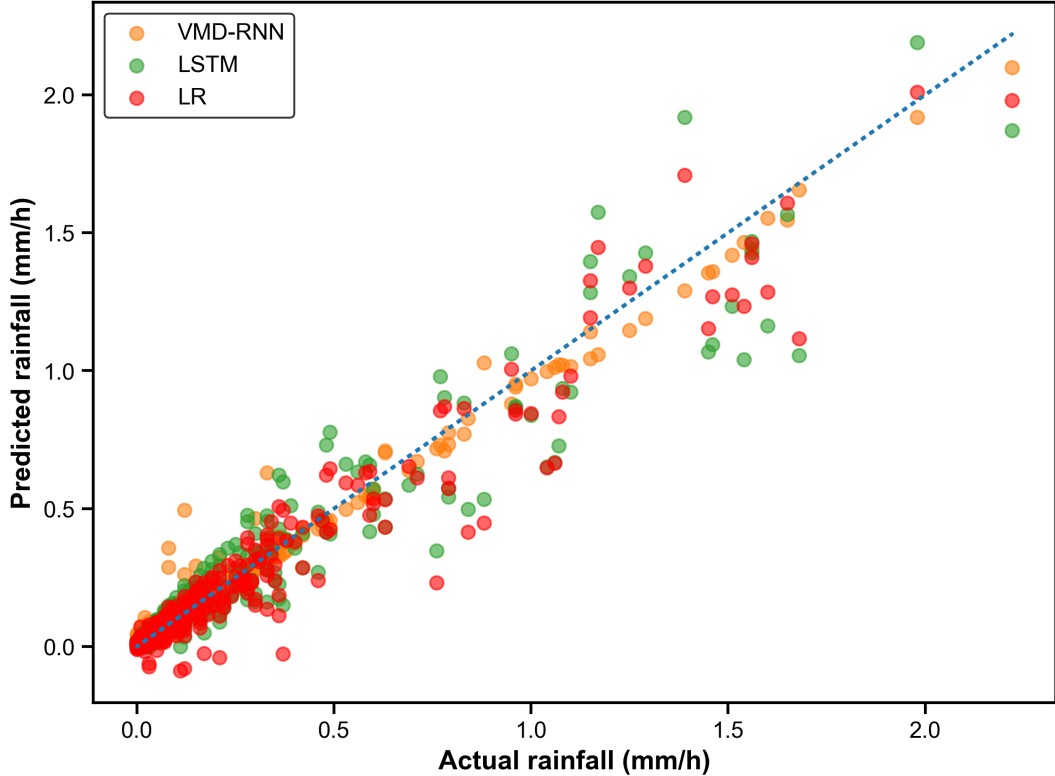

**Figure 12.** The comparison between predicted and actual hourly rainfall values

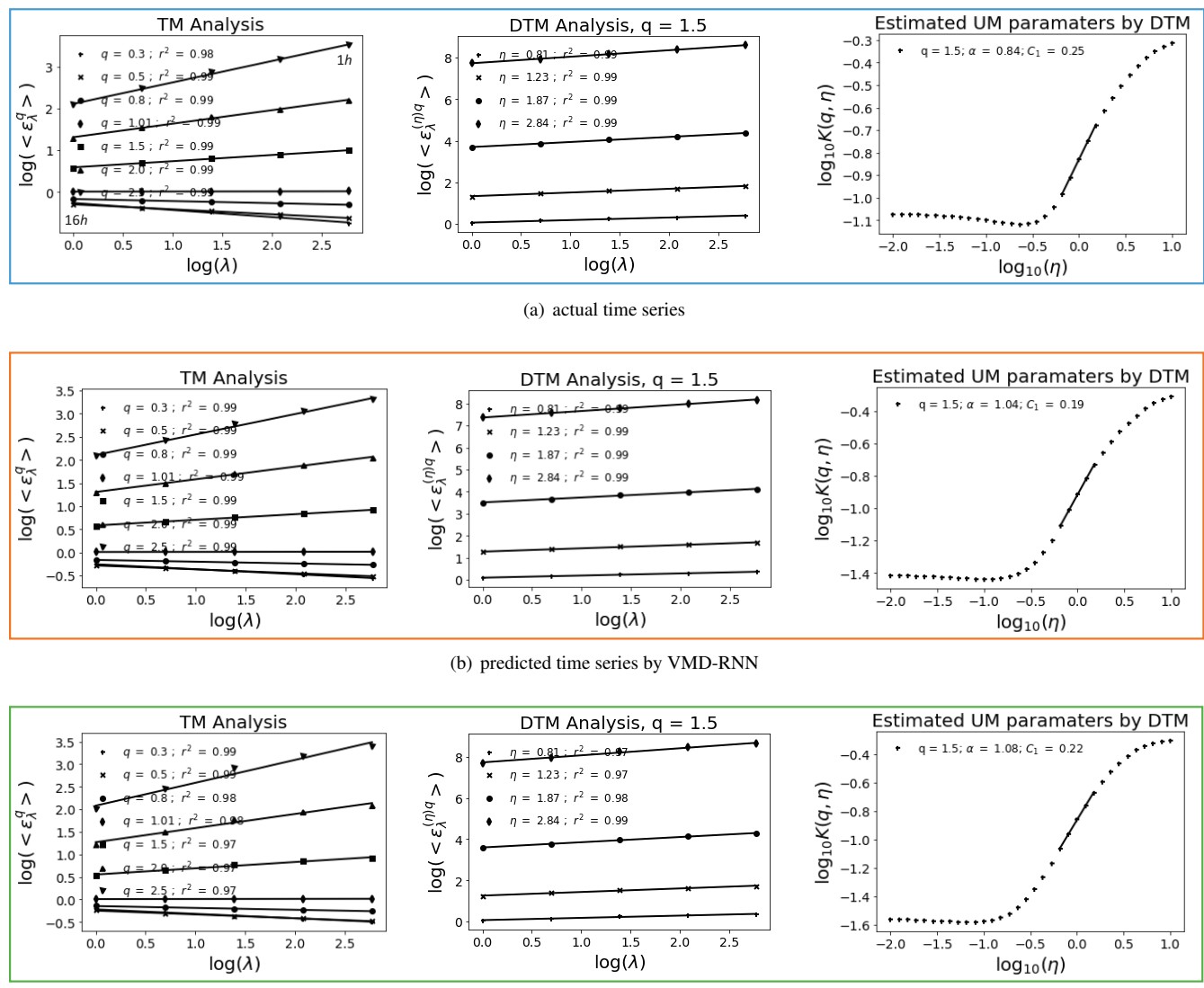

(a) actual time series

(b) predicted time series by VMD-RNN

(c) predicted time series by LSTM without decomposition

**Figure 13.** UM results for hourly time series in the testing set

**List of Tables**

555

560

**Table 1.** Relevant studies on time series prediction using deep learning models

| Reference<br>Evaluation methods | Models | Applications |
|---|---|---|
| (Ma et al., 2015)<br>MAPE, MSE | LSTM | traffic speed |
| (Ding et al., 2019)<br>RMSE, MAE | GRU | wind power |
| (Gauch et al., 2021)<br>NSE | multi-timescale LSTM | daily and hourly rainfall-runoff |
| (Ni et al., 2020)<br>RMSE, NSE, MARE | wavelet-LSTM, convolutional LSTM | monthly streamflow and rainfall |
| (Barrera-Animas et al., 2022)<br>RMSE, MAE, RMSLE | Stacked-LSTM, Bidirectional-LSTM | hourly rainfall time series |
| (He et al., 2022)<br>RMSE, NSE, MAE, Accuracy | STL-ML | daily rainfall time series |
| (Hadi and Tombul, 2018)<br>RMSE, NSE | ANN with wavelet transformation | daily streamflow |
| (Devi et al., 2020)<br>MAE, RMSE, MAPE, MASE | EEMD-CSO-LSTM-EFG | hourly wind power |
| (He et al., 2019)<br>MAE, RMSE, NSE | VMD-DNN | daily runoff |
| (Xie et al., 2019)<br>MAE, RMSE, NSE | VMD-DBN-IPSO | daily runoff series |
| (Zuo et al., 2020)<br>NSE, NRMSE, PPTS | VMD-LSTM | daily streamflow |
| this study[*]<br>RMSE, MAE, MAPE, UM | VMD-RNN | daily and hourly rainfall |

[*] This study incorporates four RNN models, namely LSTM, GRU, Bidirectional LSTM, and Bidirectional GRU. The RNN model with superior architecture was selected for each subsequence.

**Table 2.** Results of the VMD-RNN model with one hidden layer for IMF1 predicting

| Model type | Numbers of input | Model structure | MAE | RMSE | Model type | Numbers of input | Model structure | MAE | RMSE |
|---|---|---|---|---|---|---|---|---|---|
| LSTM | 5 | 32 | 0.246 | 0.496 | GRU | **5** | 32 | 0.144 | 0.380 |
| | | 64 | 0.157 | 0.396 | | | 64 | 0.150 | 0.387 |
| | | 128 | 0.188 | 0.433 | | | **128** | **0.136** | **0.369** |
| | 10 | 32 | 0.144 | 0.380 | | 10 | 32 | 0.142 | 0.377 |
| | | 64 | 0.144 | 0.380 | | | 64 | 0.171 | 0.413 |
| | | 128 | 0.174 | 0.417 | | | 128 | 0.190 | 0.436 |
| | 15 | 32 | 0.190 | 0.435 | | 15 | 32 | 0.144 | 0.379 |
| | | 64 | 0.176 | 0.420 | | | 64 | 0.163 | 0.404 |
| | | 128 | 0.160 | 0.400 | | | 128 | 0.154 | 0.392 |
| BiLSTM | 5 | 32 | 0.178 | 0.422 | BiGRU | 5 | 32 | 0.137 | 0.370 |
| | | 64 | 0.160 | 0.400 | | | 64 | 0.168 | 0.409 |
| | | 128 | 0.239 | 0.489 | | | 128 | 0.192 | 0.438 |
| | 10 | 32 | 0.158 | 0.397 | | 10 | 32 | 0.161 | 0.401 |
| | | 64 | 0.234 | 0.484 | | | 64 | 0.156 | 0.395 |
| | | 128 | 0.185 | 0.431 | | | 128 | 0.162 | 0.403 |
| | 15 | 32 | 0.138 | 0.371 | | 15 | 32 | 0.155 | 0.393 |
| | | 64 | 0.183 | 0.428 | | | 64 | 0.171 | 0.414 |
| | | 128 | 0.198 | 0.445 | | | 128 | 0.185 | 0.430 |

**Table 3.** Results of the optimal model with second and third hidden layers for IMF1 predicting

| Model type | Model structure | MAE | RMSE |
|---|---|---|---|
| GRU | 128-32 | 0.139 | 0.373 |
| | 128-64 | 0.157 | 0.396 |
| | **128-128** | **0.128** | **0.358** |
| | 128-128-32 | 0.152 | 0.39 |
| | 128-128-64 | 0.157 | 0.396 |
| | 128-128-128 | 0.17 | 0.142 |

**Table 4.** Variant RNN models of IMF1-IMF8

| VMD component | Model type | Numbers of input | Model structure |
|---|---|---|---|
| IMF1 | GRU | 5 | 128-128 |
| IMF2 | BiLSTM | 15 | 64 |
| IMF3 | BiGRU | 15 | 64-64-64 |
| IMF4 | LSTM | 10 | 64 |
| IMF5 | LSTM | 10 | 64-64-64 |
| IMF6 | BiLSTM | 15 | 64 |
| IMF7 | BiLSTM | 10 | 128-128 |
| IMF8 | BiGRU | 15 | 32-32 |

**Table 5.** Prediction errors for daily time series in the testing set

|         | MAE   | RMSE  | MAPE   |
|---------|-------|-------|--------|
| VMD-RNN | 0.726 | 0.852 | 9.853  |
| LSTM    | 6.825 | 2.612 | 10.475 |
| LR      | 9.239 | 3.040 | 18.923 |

**Table 6.** Estimated UM parameters for daily time series in the testing set

|         | TM | | DTM | |
|---------|----------|----------|----------|----------|
|         | $\alpha$ | $C_1$ | $\alpha$ | $C_1$ |
| Actual  | 0.89 | 0.25 | 1.02 | 0.24 |
| VMD-RNN | 0.98 | 0.16 | 1.06 | 0.16 |
| LSTM    | 1.11 | 0.17 | 1.23 | 0.16 |

**Table 7.** Estimated UM parameters for hourly time series in the testing set

|          | TM       |        | DTM      |        |
|----------|----------|--------|----------|--------|
|          | $\alpha$ | $C_1$  | $\alpha$ | $C_1$  |
| Actual   | 0.55     | 0.26   | 0.84     | 0.25   |
| VMD-RNN  | 0.79     | 0.21   | 1.04     | 0.19   |
| LSTM     | 0.97     | 0.22   | 1.08     | 0.22   |