# Peer review of "Combining Recurrent Neural Networks with Variational Mode Decomposition and Multifractals to Predict Rainfall Time Series"

_EGUsphere, 2023_

## Author Comment (AC2)

Thank you for your careful analysis of our manuscript and your detailed feedback. In response to your precise suggestions and comments, the corresponding response is provided as follows.

1.  *Lines 47 and 48: "However, these pure variant models are…preprocessing. " Please consider citing some previous studies to support your statement.*

We agree on the importance of backing up our statements by citing previous studies and emphasise the originality of our contributions since they aim to overcome current limitations by proposing an original combination of ML with Variational Mode Decomposition and multifractals. The preceding paragraph of this sentence (Line 47-48) has discussed these pure variant models. However in order to make it clear, we will cite these previous studies again.

2.  *The second to last and third to last paragraphs in the Introduction section should be in the Method section. They went into details about either a model or an evaluation index, rather than focusing on the context and motivation of this study.*

The suggestion regarding the placement of two paragraphs, currently in the Introduction, is duly noted. They were positioned in the Introduction because these two paragraphs describe how our work differs from others' studies and clarify the contribution of our work. We will therefore split their content between the  Introduction (more on motivations) and the Methods (more on the details) sections.

3.  *I suggest that the authors elaborate on their motivation and clarify the contribution of this work. Based on the current introduction, the model is not new, and the dataset is not new. It's okay if this work is focused on applying a method to a dataset and this application has not been documented in previous research. But you will need to justify your decision with appropriate citations. For instance, why is that application important? It could be because of limitations from previous approaches or the good performance of some new approaches, and so on. You just need to justify this work by elaborating why it is important.*

We appreciate your suggestion to elaborate more on the motivation and contribution of our work. As replied to the second comment, we will improve the introduction section by providing a more comprehensive description to emphasise the importance of our work, although we have explained our work is different and meaningful in the Section 3.1. We will clarify again in Introduction that it is mainly due to overcoming limitations of current applications of ML to rain forecasting by combining ML with Variational Mode Decomposition and Multifractals.

4.  *Maybe I missed something, but why do you need steps 3 and 4? I suggest that the authors explain why they want to generate sub-sequences on combined sequences with both training and non-training sequences and then clip to get the non-training ones instead of generating the non-training sub-sequences using directly the non-training original sequences.*

We thank you for bringing up this point. Steps 3 and 4 are included in our method due to the fact that directly decomposing the non-training original sequences will result in the leakage of future data from the testing set. Because rainfall time series is observed daily or hourly, the decomposition process is repeated with daily or hourly rainfall data of the next step appended. This approach can mitigate the risk

of exposing future data during the decomposition of non-training time series. We will add part of this discussion to clarify the methodology.

5. *Sub-section 3.3 open sources. The title of this sub-section is weird to me. Maybe consider using titles like Model Settings and Implementation*

The comment about the title of subsection 3.3 is taken into account. The subsection primarily introduces the open-source software used in this study. The suggested title 'Model Settings and Implementation' seems to be in the good direction, despite we do not implement a model in the classical sense, but set together different open-access softwares.

6. *Result analysis. Since for each testing sub-sequence several RNN models were used and only the best result was kept for result aggregation, it will be really helpful to add a summary table showing the result of each RNN model on each sub-sequence. This will not only allow readers to understand how the eventual result was aggregated but will also bring insights into which model is the best, and so on.*

It's totally agreed that your suggestion to add a summary table showing the results of each RNN model on each sub-sequence. We will include a summary table as you suggested to improve the clarity and interpretability of our results.

7. *I feel that the Result section is not very well elaborated. So far there are only results but no discussion, which damaged the value of this study. How will readers benefit from reading this paper? To me what's more important is the insights behind specific results. For instance, why are some models better than others? In what circumstances? What insights can I gain regarding model selection and tuning after reading this work? etc. I suggest the authors add more in-depth discussions (please also refer to my 6th comment) to improve the quality of this section.*

We also agree with your feedback on the Result section. We will therefore strive to provide in-depth discussions to explain the significance of our model and the contribution of our work in the field of hydrology.

---

## Author Response (AR1)

On behalf of my co-authors, we are very grateful to your affirmative recommendation on the manuscript (Manuscript Number: EGUSPHERE-2023-2710) submitted to 'Hydrology and Earth System Sciences'.

We highly appreciate your insightful comments. We have carefully read the reviewers report, and we have made corrections in the revised manuscript accordingly. Here are our responses to each of the questions in the reviewers' report, which are reproduced in italics. The newly added text has been marked in blue.

**Response to reviewer #1:**

1. *The authors have used a hybrid deep-learning model to attempt to predict rainfall. While the paper is scientifically sound and quite in-depth, I can't see it as a good article for HESS because the focus is just so strongly on the Deep-Learning infrastructure. This is already evident in the Introduction which is clearly written with an audience in mind that is up-to-date with the terminology and typical issues that come with deep-learning models, whereas there is very little attention for the real-world practical problems this model is trying to solve (the abstract mentions urban runoff issues which are never mentioned anywhere in the main body, for instance). This would be fine for a journal that focuses on that particular research area, but the typical HESS reader (or at the very least, myself) will be completely lost in the methodology section.*

We noted your concern about the manuscript's focus on deep learning models. In fact, this focus is consistent with the fact that deep learning is increasingly used in hydrological research and we aim to bridge the knowledge gap for readers who are not familiar with deep learning models. We believe that by framing the deep learning approach as a powerful tool for addressing a hydrological challenge, i.e. nowcasting, the paper becomes more attractive to a wider readership. Precipitation is so difficult to predict that one-step-ahead prediction cannot be considered to be outside of the practical problems of the real world. The following paragraph has been included to explain its suitability for HESS.

Given the growing usage of deep learning in hydrological research, it is important to bridge the knowledge gap for readers who are not familiar with deep learning models. The pedagogical aspect of our work has the potential to contribute to the hydrology community by providing a deeper understanding of the application of deep learning models and multifractals technique in short-term rainfall prediction that remains a fundamental problem of hydrology starting with one-step-ahead prediction.

2. *It's way too detailed in explaining the core mathematical concepts behind the model (once again, scientifically absolutely good work, but not for a hydrology-focused journal), whereas the section discussing the used dataset for validation purposes (section 3.1) is barely 10 lines long and doesn't contain any information about the type of data collected (is it radar, tipping bucket, time-integrated, point measurements, etc etc).*

We greatly simplified the mathematical presentation, e.g., we removed two paragraphs in the introduction that describe the detailed implementation and concepts of the hybrid model. We also simplified the description about double trace moment technique in section 2.3.

We provide some information on the dataset we used for training in section 3.1, e.g.:

Two rainfall time series with daily and hourly resolutions in Champs-sur-Marne ($48.8425°N$, $2.5886°E$) were collected from MERRA-2 (Modern-Era Retrospective analysis for Research and Applications, Version 2) precipitation dataset that is produced by NASA's Global Modeling and Assimilation Office (GMAO), refer to The POWER Project (https://power.larc.nasa.gov). The corrected MERRA-2 precipitation dataset is a reanalysis product that integrates various observational data types (like radar, tipping bucket gauges, and satellite) through sophisticated data assimilation techniques into a climate model.

We also clarify that the role of the training set is less stringent than believed at first glance and therefore the transportability of the model is much greater: One could worry about the model's applicability beyond the chosen study area , i.e. its transportability,  because the model only has to be trained once. In principle, a new dataset from different regions or time periods can be fed directly into the well-trained model without repeating the training process to obtain the prediction on the new dataset.

3. *That aside, the outcomes of the study are also a bit disappointing from a practical point of view. The authors acknowledge that their chosen study area has a fairly typical rain pattern, which makes me wonder whether this means such a model can't be applied anywhere else without specifically training it for that area - which would defeat the purpose of using a model, in my opinion.*

*We have just provided an answer to this concern*

*Secondly, and perhaps most importantly: the authors conclude that with the used lead time (1 time step) the applicability of the model is severely limited for prediction purposes, nor can it handle the stochastic nature of rainfall variability all too well. A conclusion on my end would be then that it's not any better than just interpolating observational data...*

We noted your opinion about the limited attention to real-world practical problems. However, we already pointed out that precipitation is so difficult to predict that one-step-ahead prediction cannot be considered outside the practical problems of the real world. We believe that this point of view would be supported by urban water managers. Furthermore, the study presented in this manuscript serves as a kind of pedagogical example, acting as a starting point for further research that can extend to longer lead-time nowcasting. As we described in the future works, multi-step-ahead rainfall prediction is currently under investigation, and the model combined multifractals with deep learning is being developed to analyse and monitor the variability of forecast rainfall time series.

To clarify that our deep learning prediction cannot be considered as similar to a linear interpolation we have introduced the traditional linear regression method as one of baseline methods in the Section 4. Figures 8 and 9 for daily resolution, and Figures 11 and 12 for hourly resolution have updated to include the results of the linear regression method, which do not fit deep learning prediction. We include below copies of Figs 8-12.

[Figure]

Figures 8: Predicted and actual daily time series in the testing set

[Figure]

Figures 9: The comparison between predicted and actual daily rainfall values

[Figure]

Figures 11: Predicted and actual hourly time series in the testing set

[Figure]

Figures 12: The comparison between predicted and actual hourly rainfall values

**Response to reviewer #2:**

1. *Lines 47 and 48: "However, these pure variant models are…preprocessing." Please consider citing some previous studies to support your statement.*

We agree on the importance of backing up our statements by citing previous studies and emphasise the originality of our contributions since they aim to overcome current limitations by employing the combination of DL with Decomposition. Therefore, five references (Liu et al., 2020; Huang et al., 2021; Zhang et al., 2021; Lv and Wang, 2022; Ruan et al., 2022) have been cited to support the statement.

2. *The second to last and third to last paragraphs in the Introduction section should be in the Method section. They went into details about either a model or an evaluation index, rather than focusing on the context and motivation of this study.*

The suggestion regarding the placement of two paragraphs, currently in the Introduction, is duly taken into account. They were initially positioned in the Introduction because these two paragraphs describe how our work differs from others' studies and clarify the contribution of our work. The three following paragraphs havereplaced the two original paragraphs to explain the purpose and motivation of this study.

The inherent variability of rainfall typically results in limited prediction performance for single RNN-variant models. In response to this situation, the integrated forecasting paradigms have been widely employed to improve the precision and robustness of time series forecasting. The hybrid VMD-RNN model is based on the fundamental concept of considering the dominant characteristics of VMD in decomposing nonlinear time series and the beneficial performance of variant RNN models in predicting complex sequential problems.

The main purpose of this study is to provide a reliable one-step-ahead rainfall prediction. In order to achieve this objective, it is essential to fully extract the underlying patterns of rainfall time series. An additional crucial point is to develop prediction models with a satisfactory level of accuracy. According to the aforementioned two factors, this study implements a hybrid approach known as VMD-RNN, which combines different RNN-variant models with VMD decomposition for predicting rainfall time series.

The effectiveness and reliability of the employed VMD-RNN approach are extensively validated by applying this method to forecast the following step's rainfall in both daily and hourly resolution. Furthermore, a comparison study is carried out to further demonstrate the superiority of the adopted VMD-RNN model, in comparison to the baseline method, pure LSTM model without decomposition. In addition, the UM technique is used to confirm the ability of the predicted time series to accurately describe rainfall variability.

3. *I suggest that the authors elaborate on their motivation and clarify the contribution of this work. Based on the current introduction, the model is not new, and the dataset is not new. It's okay if this work is focused on applying a method to a dataset and this application has not been documented in previous research. But you will need to justify your decision with appropriate citations. For instance, why is that application important? It could be because of limitations from previous approaches or the good performance of some new approaches, and so on. You just need to justify this work by elaborating why it is important.*

We appreciate your suggestion to elaborate more on the motivation and contribution of our work. As replied to the second comment, we improved the introduction section by providing a more comprehensive description to emphasize the importance of our work, e.g., the number of decomposition levels in the process of VMD is determined by analyzing the power spectral density of the corresponding last sub-sequence.

As responded to the second comment, the advantages and the purposes of combining RNN models with variational mode decomposition and multifractals have been explained in the three new paragraphs in the Introduction section.

4. *Maybe I missed something, but why do you need steps 3 and 4? I suggest that the authors explain why they want to generate sub-sequences on combined sequences with both training and non-training sequences and then clip to get the non-training ones instead of generating the non-training sub-sequences using directly the non-training original sequences.*

We thank you for bringing up this point. Steps 3 and 4 are included in our method due to the fact that directly decomposing the non-training original sequences will result in the leakage of future data from the testing set. Because rainfall time series is observed daily or hourly, the decomposition process is repeated with daily or hourly rainfall data of the next step appended. This approach can mitigate the risk of exposing future data during the decomposition of non-training time series. The following paragraph has been added to clarify the methodology.

To minimize the possibility of exposing future data during the decomposition of non-training time series, a precautionary approach (Step 3 and Step 4) has been implemented. This approach differs from the direct way of decomposing the testing time series using VMD. The non-training data was added to the training set in a sequential manner to create a new time series, and the amount of new generated time series was equal to the number of non-training data points. The VMD technique was thereafter used to decompose the aforementioned new time series into several sub-sequences. Subsequently, the final data point of each newly generated sub-sequence was retrieved and designated as non-training data, which was then used to build validation and testing samples.

5. *Sub-section 3.3 open sources. The title of this sub-section is weird to me. Maybe consider using titles like Model Settings and Implementation*

The comment about the title of subsection 3.3 is taken into account. The subsection primarily introduces the open-source software used in this study. The suggested title 'Model Settings and Implementation' seems to be in the good direction, despite we do not implement a model in the classical sense, but set together different open-access software. The title has been changed to 'Open-source software'

*6. Result analysis. Since for each testing sub-sequence several RNN models were used and only the best result was kept for result aggregation, it will be really helpful to add a summary table showing the result of each RNN model on each sub-sequence. This will not only allow readers to understand how the eventual result was aggregated but will also bring insights into which model is the best, and so on.*

We totally agreed with your suggestion to add summary tables showing the results of each RNN model on each sub-sequence. Table 2 and Table 3 have been included in the section 3.2.3 to improve the clarity and interpretability of our results. Two tables show the MAE and RMSE results of the optimal RNN-variant model with first, second and third hidden layers for predicting first sub-sequence (IMF1). Then, Table 4 succinctly presents the ideal models with optimal parameters for other sub-sequences, which were obtained by the same way as IMF1.

**Table 2.** Results of the VMD-RNN model with one hidden layer for IMF1 predicting

| Model type | Numbers of input | Model structure | MAE | RMSE | Model type | Numbers of input | Model structure | MAE | RMSE |
|---|---|---|---|---|---|---|---|---|---|
| | | 32 | 0.246 | 0.496 | | | 32 | 0.144 | 0.380 |
| | 5 | 64 | 0.157 | 0.396 | | 5 | 64 | 0.150 | 0.387 |
| | | 128 | 0.188 | 0.433 | | | **128** | **0.136** | **0.369** |
| | | 32 | 0.144 | 0.380 | | | 32 | 0.142 | 0.377 |
| LSTM | 10 | 64 | 0.144 | 0.380 | GRU | 10 | 64 | 0.171 | 0.413 |
| | | 128 | 0.174 | 0.417 | | | 128 | 0.190 | 0.436 |
| | | 32 | 0.190 | 0.435 | | | 32 | 0.144 | 0.379 |
| | 15 | 64 | 0.176 | 0.420 | | 15 | 64 | 0.163 | 0.404 |
| | | 128 | 0.160 | 0.400 | | | 128 | 0.154 | 0.392 |
| | | 32 | 0.178 | 0.422 | | | 32 | 0.137 | 0.370 |
| | 5 | 64 | 0.160 | 0.400 | | 5 | 64 | 0.168 | 0.409 |
| | | 128 | 0.239 | 0.489 | | | 128 | 0.192 | 0.438 |
| | | 32 | 0.158 | 0.397 | | | 32 | 0.161 | 0.401 |
| BiLSTM | 10 | 64 | 0.234 | 0.484 | BiGRU | 10 | 64 | 0.156 | 0.395 |
| | | 128 | 0.185 | 0.431 | | | 128 | 0.162 | 0.403 |
| | | 32 | 0.138 | 0.371 | | | 32 | 0.155 | 0.393 |
| | 15 | 64 | 0.183 | 0.428 | | 15 | 64 | 0.171 | 0.414 |
| | | 128 | 0.198 | 0.445 | | | 128 | 0.185 | 0.430 |

**Table 3.** Results of the optimal model with second and third hidden layers for IMF1 predicting

| Model type | Model structure | MAE | RMSE |
|---|---|---|---|
| | 128-32 | 0.139 | 0.373 |
| | 128-64 | 0.157 | 0.396 |
| GRU | **128-128** | **0.128** | **0.358** |
| | 128-128-32 | 0.152 | 0.39 |
| | 128-128-64 | 0.157 | 0.396 |
| | 128-128-128 | 0.17 | 0.142 |

**Table 4.** Variant RNN models of IMF1-IMF8

| VMD component | Model type | Numbers of input | Model structure |
|---|---|---|---|
| IMF1 | GRU | 5 | 128-128 |
| IMF2 | BiLSTM | 15 | 64 |
| IMF3 | BiGRU | 15 | 64-64-64 |
| IMF4 | LSTM | 10 | 64 |
| IMF5 | LSTM | 10 | 64-64-64 |
| IMF6 | BiLSTM | 15 | 64 |
| IMF7 | BiLSTM | 10 | 128-128 |
| IMF8 | BiGRU | 15 | 32-32 |

*7. I feel that the Result section is not very well elaborated. So far there are only results but no discussion, which damaged the value of this study. How will readers benefit from reading this paper? To me what's more important is the insights behind specific results. For instance, why are some models better than others? In what circumstances? What insights can I gain regarding model selection and tuning after reading this work? etc. I suggest the authors add more in-depth discussions (please also refer to my 6th comment) to improve the quality of this section.*

We also agree with your feedback on the Result section. We therefore strived to provide in-depth discussions to explain the significance of our model and the contribution of our work in the field of hydrology. The following discussion has been added to further explain the results.

The hybrid VMD-RNN model, which integrates VMD decomposition and several RNN-variant models, showed a powerful ability to predict the next step's rainfall time series at both daily and hourly resolution. In order to further verify the effectiveness of the hybrid VMD-RNN approach, two baseline methods (the pure LSTM model without decomposition and the linear regression model) were also tested with the same daily and hourly rainfall time series. The hybrid VMD-RNN model and the baseline method were compared to highlight the necessity of VMD decomposition and every RNN-variant model for accurate rainfall prediction.

In terms of the regression results of daily time series, the hybrid VMD-RNN model outperforms the baseline methods in regards to the prediction of rainfall values. The findings obtained from Table 5 indicate the superiority of the used hybrid approach in daily rainfall regression, as evidenced by the lower values of MAE, RMSE, and MAPE. In addition, the scatter plot in Figure 9 shows that the baseline models consistently underestimate the intensity of rainfall, resulting in misjudgement and delayed responses to potential flood disasters. For hourly rainfall time series, the prediction performance of VMD-RNN is comparable to that of the pure LSTM model, without demonstrating substantial advantages of decomposition, which can be attributed to the small values of hourly time series.

According to the results of multifractal analysis, the UM parameters obtained from the time series predicted by VMD-RNN exhibit a higher degree of similarity to the actual time series, in comparison to the parameters from the time series predicted by LSTM without decomposition, specifically for daily time series. The values of $C_1$ calculated from predicted time series are lower, which is due to the fact that predicted time series tend to produce very small values rather than indicating the absence of rainfall. However, in the case of hourly time series, the UM results quantitatively suggest that the predictive performance of the VMD-RNN model is similar to that of the pure LSTM model, without explicitly showing the advantages of decomposition.

---

## Author Response (AR2)

Dear Editor

My co-authors and I are very grateful for your positive recommendation on the manuscript (Manuscript Number: EGUSPHERE-2023-2710) submitted to 'Hydrology and Earth System Sciences'.

We highly appreciate your insightful comments. We have carefully read the reviewers report, and we have made corrections in the revised manuscript accordingly. Here are our responses to each of the questions in the reviewers' report, which are reproduced in italics. The newly added text has been marked in blue.

**Response to reviewer #1:**

1. *While the authors have made commendable efforts to make their paper more accessible for HESS, based on the earlier reviews, my initial doubts from the first revision about whether this article fits the scope of HESS are not really taken away. Now the authors mention in L90-95 the 'pedagogical aspect' of the work, but that also requires some form of relatedness to the field of the learner (the HESS community). The authors mention the "fundamental problem [...] starting with one-step ahead prediction" and then give no context for this problem, nor any mention of nowcasting studies, or any physics-based rainfall model study.*

We have noted your concern regarding the manuscript's suitability for HESS. To address this, we have revised the following paragraphs to the introduction to better frame our work in the context of hydrological applications and to emphasize its pedagogical value for the HESS community.

The main purpose of this study is to provide a reliable one-step-ahead rainfall prediction for hydrological applications, particularly urban flood forecasting and water resource management. This addresses the fundamental challenge in operational hydrology where accurate short-term precipitation forecasts are essential for timely flood warnings and infrastructure management. In order to achieve this objective, it is essential to fully extract the underlying patterns of rainfall time series while preserving their multiscale intermittency structure - a critical requirement for hydrological modeling where extreme events often dominate system response. An additional crucial point is to develop prediction models with a satisfactory level of accuracy for practical implementation in operational hydrological systems. According to the aforementioned two factors, this study implements a hybrid approach known as VMD-RNN, which combines different Recurrent Networks models (RNN) with Variational Mode Decomposition (VMD) to take into account the multiscale nature of precipitation. Moreover, a scaling technique is used to optimize the width of decomposition.

The effectiveness and reliability of the employed VMD-RNN approach are extensively validated by applying this method to forecast the following step's rainfall in both daily and hourly resolution, representing different temporal scales relevant to hydrological practice.

Furthermore, a comparison study is carried out to further demonstrate the superiority of the adopted VMD-RNN model, in comparison to the baseline method, the pure Long Short-Term Memory model (LSTM) model without decomposition, and linear regression method. In addition, a Universal Multifractal (UM) technique is used to confirm the ability of the predicted time series to accurately describe rainfall variability, ensuring that the predicted series maintain the multifractal properties essential for accurate hydrological modeling and flood risk assessment.

Given the growing usage of deep learning in hydrological research, it is important to bridge the knowledge gap for readers who are not familiar with deep learning models. The pedagogical aspect of our work has the potential to contribute to the hydrology community by providing a deeper understanding of the application of deep learning models and multifractal techniques in short-term rainfall prediction that remains a fundamental problem of hydrology starting with one-step-ahead prediction. This work specifically addresses the need in the HESS community for accessible methodological advances that maintain strong connections to hydrological theory and practice, demonstrating how modern deep learning techniques can enhance traditional approaches to precipitation forecasting while preserving the physical understanding of rainfall processes essential for water resource management.

2. *The Results section is still very meager (though I do appreciate the inclusion of the dataset details now), and while figures 11-13 give some measure of improvement over LSTM and a benchmark linear regression (the authors also don't mention the LR parameters), there is still some very large errors especially at the lowest rainfall values (<4 mm/hr) where there are either False Zeroes or High Influx related errors. This is not discussed anywhere, whereas that results discussion is crucial for any form of applicability: does the model perform poorly at low intensity so is it only useful for applications near the high-end of rainfall distributions, are there ways this can be alleviated, etc.*

We agree with your assessment that the Results section required a more detailed discussion, particularly concerning the model's performance at different rainfall intensities. We have substantially revised the Results section to provide a more in-depth analysis of the model's performance for both daily and hourly rainfall, including a discussion of the errors at low rainfall values.

4.1 Daily rainfall series
Figure 8 shows the predicted daily time series in the testing set. It compares the predicted results of the VMD-RNN hybrid model, the pure LSTM model and the linear regression method with the actual data. It can be clearly observed that the hybrid model has a better fit for most of the points, particularly during periods of high-intensity rainfall events that are critical for flood forecasting applications. The VMD-RNN model demonstrates enhanced capability to capture rainfall variability patterns, including the temporal clustering of precipitation events that characterizes real rainfall processes.

The comparison of prediction performance with and without VMD for daily time series in the testing set can be seen in Figure 9. The scatter plot demonstrates that the VMD-RNN model has superior performance in predicting both high and low values for daily time series, whereas the baseline models LSTM and linear regression exhibit systematic biases. Notably, the VMD-RNN model shows improved performance in predicting extreme rainfall events, which are crucial for urban flood warning systems. The predicted values obtained by the baseline models exhibit considerable deviation from the best linear fitting line (blue dotted line), with a tendency to underestimate high-intensity events - a critical limitation for hydrological applications where accurate prediction of extreme events directly impacts flood risk assessment and emergency response effectiveness.

All the parameter values estimated using the TM and DTM methods are listed in Table 6. The values of $\alpha$ and $C_1$ obtained using the DTM technique show slight differences from those estimated by TM, but remain within acceptable ranges for multifractal analysis. Importantly, the VMD-RNN predicted time series preserves multifractal properties more effectively than LSTM without decomposition, as evidenced by UM parameters that are closer to those of actual rainfall. This preservation of scaling properties is crucial for hydrological applications where the multifractal structure of rainfall directly influences runoff generation, infiltration processes, and the temporal distribution of streamflow in urban catchments.

4.2 Hourly rainfall series

Figure 11 displays the hourly time series in the testing set with 1024 data points. The qualitative analysis reveals that the predictive performance differences between VMD-RNN, pure LSTM, and linear regression are less pronounced for hourly rainfall time series compared to daily predictions. This reduced benefit of decomposition for hourly data can be attributed to the inherently higher noise level and lower signal-to-noise ratio characteristic of high-frequency precipitation measurements, which limits the effectiveness of decomposition techniques in extracting meaningful frequency components.

Figure 12 depicts the comparison between predicted and actual hourly rainfall values. The scatter plot reveals that the predicted values from VMD-RNN basically agree with the corresponding actual values, but the values predicted from the baseline LSTM model do not yield the same level of alignment. While the VMD-RNN model shows reasonable agreement with actual values for moderate to high rainfall intensities, significant challenges become apparent for low-intensity precipitation events.

The UM analysis results for hourly time series (Figure 13) and estimated parameters (Table 7) indicate that the predictive performance of VMD-RNN is comparable to pure LSTM for hourly data, without demonstrating the substantial benefits observed for daily predictions. The UM parameters $\alpha$ and $C_1$ show similar values between VMD-RNN and LSTM predictions, suggesting that both approaches preserve multifractal properties to a similar degree at hourly resolution. This finding reflects the scale-dependent effectiveness of

decomposition techniques, where the benefits become more apparent at longer timescales where signal-to-noise ratios are higher and frequency separation is more pronounced.

3. *Additionally, the figure captions seem to be missing or are in any case in complete; figure quality is also quite poor, and in figure 12 there are no axis labels so indicate what is being displayed.*

We apologize for the issues with the figures. We have improved the quality of all figures, added the missing axis labels, and ensured all captions are complete and descriptive.

[Figure]

Figure 9. The comparison between predicted and actual daily rainfall values

[Figure]

Figure 12. The comparison between predicted and actual hourly rainfall values

**Response to editor:**

1. *The latter mainly indicates that the paper's results are meager (I agree but on the other hand, the paper makes one point and does so with clear focus. So that is fine for me.*

Thank you for your comment. We are pleased that you found the focus of the paper to be clear. At the same time, we hope that the current modifications make more obvious the results and their importance.

2. *The second point however, is that the paper is not that accessible to the Hess community (being hydrologists and hydrometeorologists). If I read the discussion and conclusion sections, I do see (and agree with) that point. The discussion, for example, has no references, no broader context. It is more like three paragraphs technical summary of the work. This broader (hydrological application) discussion of your results would be of added value. This then also should tickle down in the conclusions section.*

We agree that the Discussion section needed to be expanded to provide broader context and better connect our findings with the hydrology community. We have completely rewritten the Discussion section (now Section 4.3) to address this, including a comparison with existing approaches, a detailed explanation of the hydrological significance, and a discussion of the model's limitations.

[revised manuscript text omitted]

3. *Please also provide higher quality figures.*

We have replaced all figures with high-resolution versions as requested, particularly Figures 9 and 12.